# Beyond Perturbations: Learning Guarantees with Arbitrary Adversarial Test Examples

**Shafi Goldwasser***
UC Berkeley, MIT

**Adam Tauman Kalai**
Microsoft Research

**Yael Tauman Kalai**
Microsoft Research, MIT

**Omar Montasser**
TTI Chicago

## Abstract

We present a transductive learning algorithm that takes as input training examples from a distribution $P$ and *arbitrary* (unlabeled) test examples, possibly chosen by an adversary. This is unlike prior work that assumes that test examples are small perturbations of $P$. Our algorithm outputs a *selective classifier*, which abstains from predicting on some examples. By considering selective transductive learning, we give the first nontrivial guarantees for learning classes of bounded VC dimension with arbitrary train and test distributions—no prior guarantees were known even for simple classes of functions such as intervals on the line. In particular, for any function in a class $C$ of bounded VC dimension, we guarantee a low test error rate and a low rejection rate *with respect to* $P$. Our algorithm is efficient given an Empirical Risk Minimizer (ERM) for $C$. Our guarantees hold even for test examples chosen by an unbounded white-box adversary. We also give guarantees for generalization, agnostic, and unsupervised settings.

## 1 Introduction

Consider binary classification where test examples are not from the training distribution. Specifically, consider learning a binary function $f : X \rightarrow \{0, 1\}$ where training examples are assumed to be iid from a distribution $P$ over $X$, while the test examples are *arbitrary*. This includes both the possibility that test examples are chosen by an adversary or that they are drawn from a distribution $Q \neq P$ (sometimes called "covariate shift"). For a disturbing example of covariate shift, consider learning to classify abnormal lung scans. A system trained on scans prior to 2019 may miss abnormalities due to COVID-19 since there were none in the training data. As a troubling adversarial example, consider explicit content detectors which are trained to classify normal vs. explicit images. Adversarial spammers synthesize endless variations of explicit images that evade these detectors for purposes such as advertising and phishing [Yuan et al., 2019].

A recent line of work on adversarial learning has designed algorithms that are robust to imperceptible perturbations. However, perturbations do not cover all types of test examples. In the explicit image detection example, Yuan et al. [2019] find adversaries using conspicuous image distortion techniques (e.g., overlaying a large colored rectangle on an image) rather than imperceptible perturbations. In the lung scan example, Fang et al. [2020] find noticeable signs of COVID in many scans.

In general, there are several reasons why learning with arbitrary test examples is actually impossible. First of all, one may not be able to predict the labels of test examples that are far from training examples, as illustrated by the examples in group (1) of Figure 1. Secondly, as illustrated by group (2), given any classifier $h$, an adversary or test distribution $Q$ may concentrate on or near an error. High error rates are thus unavoidable since an adversary can simply repeat any single erroneous example they can find. This could also arise naturally, as in the COVID example, if $Q$ contains a concentration of new examples near one another–individually they appear "normal" (but are suspicious as a group).

This is true even under the standard *realizable* assumption that the target function $f \in C$ is in a known class $C$ of bounded VC dimension $d = VC(C)$.

As we now argue, learning with arbitrary test examples requires *selective classifiers* and *transductive learning*, which have each been independently studied extensively. We refer to the combination as classification with *redaction*, a term which refers to the removal/obscuring of certain information when documents are released. A *selective classifier* (SC) is one which is allowed to abstain from predicting on some examples. In particular, it specifies both a classifier $h$ and a subset $S \subseteq X$ of examples to classify, and rejects the rest. Equivalently, one can think of a SC as $h|_S : X \to \{0, 1, \blacksquare\}$ where $\blacksquare$ indicates $x \notin S$, abstinence.

$$h|_S(x) := \begin{cases} h(x) & \text{if } x \in S \\ \blacksquare & \text{if } x \notin S. \end{cases}$$

We say the learner *classifies* $x$ if $x \in S$ and otherwise it rejects $x$. Following standard terminology, if $x \notin S$ (i.e., $h|_S(x) = \blacksquare$) we say the classifier *rejects* $x$ (the term is not meant to indicate anything negative about the example $x$ but merely that its classification may be unreliable). We sat that $h|_S$ *misclassifies* or *errs* on $x$ if $h|_S(x) = 1 - f(x)$. There is a long literature on SCs, starting with the work of Chow [1957] on character recognition. In standard classification, *transductive learning* refers to the simple learning setting where the goal is to classify a given unlabeled test set that is presented together with the training examples [see e.g., Vapnik, 1998]. We will also consider the generalization error of the learned classifier.

This raises the question: *When are unlabeled test examples available in advance?* In some applications, test examples are classified all at once (or in batches). Otherwise, redaction can also be beneficial *in retrospect*. For instance, even if image classifications are necessary immediately, an offensive image detector may be run daily with rejections flagged for inspection; and images may later be blocked if they are deemed offensive. Similarly, if a group of unusual lung scans showing COVID were detected after a period of time, the recognition of the new disease could be valuable even in hindsight. Furthermore, in some applications, one cannot simply label a sample of test examples. For instance, in learning to classify messages on an online platform, test data may contain both public and private data while training data may consist only of public messages. Due to privacy concerns, labeling data from the actual test distribution may be prohibited.

It is clear that a SC is necessary to guarantee few test misclassifications, e.g., if $P$ is concentrated on a single point $x$, rejection is necessary to guarantee few errors on arbitrary test points. However, no prior guarantees (even statistical guarantees) were known even for learning elementary classes such as intervals or halfspaces with arbitrary $P \neq Q$. This is because learning such classes is impossible without unlabeled examples.

To illustrate how redaction (transductive SC) is useful, consider learning an interval $[a, b]$ on $X = \mathbb{R}$ with arbitrary $P \neq Q$. This is illustrated below with (blue) dots indicating test examples:

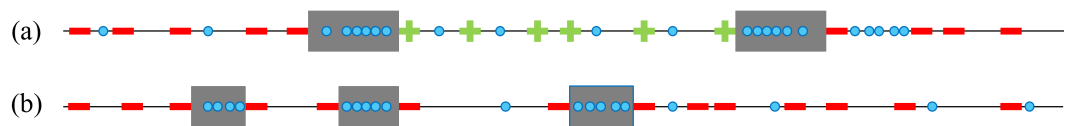

With positive training examples as in (a), one can guarantee 0 test errors by rejecting the two (grey) regions adjacent to the positive examples. When there are no positive training examples,[2] as in (b), one can guarantee $\leq k$ test errors by rejecting any region with $> k$ test examples and no training examples; and predicting negative elsewhere. Of course, one can guarantee 0 errors by rejecting everywhere, but that would mean rejecting even future examples distributed like $P$. While our error objective will be an $\epsilon$ test error rate, our rejection objective will be more subtle since we cannot absolutely bound the test rejection rate. Indeed, as illustrated above, in some cases one should reject many test examples.

Note that our redaction model assumes that the target function $f$ remains the same at train and test times. This assumption holds in several (but not all) applications of interest. For instance, in explicit

image detection, U.S. laws regarding what constitutes an illegal image are based solely on the image $x$ itself [U.S.C., 1996]. Of course, if laws change between train and test time, then $f$ itself may change. *Label shift* problems where $f$ changes from train to test is also important but not addressed here. Our focus is primarily the well-studied realizable setting, where $f \in C$, though we analyze an agnostic setting as well.

## 1.1 Redaction model and guarantees

Our goal is to learn a target function $f \in C$ of VC dimension $d$ with training distribution $P$ over $X$. In the redaction model, the learner first chooses $h \in C$ based on $n$ iid training examples $\mathbf{x} \sim X^n$ and their labels $f(\mathbf{x}) = \big(f(x_1), f(x_2), \dots, f(x_n)\big) \in \{0,1\}^n$. (In other words, it trains a standard binary classifier.) Next, a "white box" *adversary* selects $n$ arbitrary test examples $\tilde{\mathbf{x}} \in X^n$ based on all information including $\mathbf{x}, f, h, P$ and the learning algorithm. Using the unlabeled test examples (and the labeled training examples), the learner finally outputs $S \subseteq X$. Errors are those test examples in $S$ that were misclassified, i.e., $h|_S(x) = 1 - f(x)$.

Rather than jumping straight into the transductive setting, we first describe the simpler generalization setting. We define the *PQ* model in which $\tilde{\mathbf{x}} \sim Q^n$ are drawn iid by *nature*, for an arbitrary distribution $Q$. While it will be easier to quantify generalization error and rejections in this simpler model, the *PQ* model does not permit a white-box adversary to choose test examples based on $h$. To measure performance here, define rejection and error rates for distribution $D$, respectively:

$$\blacksquare_D(S) := \Pr_{x \sim D}[x \notin S] \tag{1}$$

$$\mathrm{err}_D(h|_S) := \Pr_{x \sim D}[h(x) \neq f(x) \wedge x \in S] \tag{2}$$

We write $\blacksquare_D$ and $\mathrm{err}_D$ when $h$ and $S$ are clear from context. We extend the definition of PAC learning to $P \neq Q$ as follows:

**Definition 1.1** (PQ learning). *Learner $L$ $(\epsilon, \delta, n)$-PQ-learns $C$ if for any distributions $P, Q$ over $X$ and any $f \in C$, its output $h|_S = L(\mathbf{x}, f(\mathbf{x}), \tilde{\mathbf{x}})$ satisfies*

$$\Pr_{\mathbf{x} \sim P^n, \tilde{\mathbf{x}} \sim Q^n} \big[ \blacksquare_P + \mathrm{err}_Q \leq \epsilon \big] \geq 1 - \delta.$$

*$L$ PQ-learns $C$ if $L$ runs in polynomial time and if there is a polynomial $p$ such that $L$ $(\epsilon, \delta, n)$-PQ-learns $C$ for every $\epsilon, \delta > 0, n \geq p(1/\epsilon, 1/\delta)$.*

Now, at first it may seem strange that the definition bounds $\blacksquare_P$ rather than $\blacksquare_Q$, but as mentioned $\blacksquare_Q$ cannot be bound absolutely. Instead, it can be bound relative to $\blacksquare_P$ and the *total variation distance* (also called statistical distance) $|P - Q|_{\mathrm{TV}} \in [0, 1]$, as follows:

$$\blacksquare_Q \leq \blacksquare_P + |P - Q|_{\mathrm{TV}}.$$

This new perspective, of bounding the rejection probability of $P$, as opposed to $Q$, facilitates the analysis. Of course when $P = Q$, $|P - Q|_{\mathrm{TV}} = 0$ and $\blacksquare_Q = \blacksquare_P$, and when $P$ and $Q$ have disjoint supports (no overlap), then $|P - Q|_{\mathrm{TV}} = 1$ and the above bound is vacuous. We also discuss tighter bounds relating $\blacksquare_Q$ to $\blacksquare_P$.

We provide two redactive learning algorithms: a supervised algorithm called Rejectron, and an unsupervised algorithm URejectron. Rejectron takes as input $n$ labeled training data $(\mathbf{x}, \mathbf{y}) \in X^n \times \{0,1\}^n$ and $n$ test data $\tilde{\mathbf{x}} \in X^n$ (and an error parameter $\epsilon$). It can be implemented efficiently using any $\mathrm{ERM}_C$ oracle that outputs a function $c \in C$ of minimal error on any given set of labeled examples. It is formally presented in Figure 2. At a high level, it chooses $h = \mathrm{ERM}(\mathbf{x}, \mathbf{y})$ and chooses $S$ in an iterative manner. It starts with $S = X$ and then iteratively chooses $c \in C$ that disagrees significantly with $h|_S$ on $\tilde{\mathbf{x}}$ but agrees with $h|_S$ on $\mathbf{x}$; it then rejects all $x$'s such that $c(x) \neq h(x)$. As we show in Lemma 4.1, choosing $c$ can be done efficiently given oracle access to $\mathrm{ERM}_C$.

Theorem 4.2 shows that Rejectron PQ-learns any class $C$ of bounded VC dimension $d$, specifically with $\epsilon = \tilde{O}(\sqrt{d/n})$. (The $\tilde{O}$ notation hides logarithmic factors including the dependence on the failure probability $\delta$.) This is worse than the standard $\epsilon = \tilde{O}(d/n)$ bound of supervised learning when $P = Q$, though Theorem 4.4 shows this is necessary with an $\Omega(\sqrt{d/n})$ lower-bound for $P \neq Q$.

Our unsupervised learning algorithm URejectron, formally presented in Figure 3, computes $S$ only from unlabeled training and test examples, and has similar guarantees (Theorem 4.5). The algorithm

tries to distinguish training and test examples and then rejects whatever is almost surely a test example. More specifically, as above, it chooses $S$ in an iterative manner, starting with $S = X$. It (iteratively) chooses *two* functions $c, c' \in C$ such that $c|_S$ and $c'|_S$ have high disagreement on $\tilde{\mathbf{x}}$ and low disagreement on $\mathbf{x}$, and rejects all $x$'s on which $c|_S, c'|_S$ disagree. As we show in Lemma B.1, choosing $c$ and $c'$ can be done efficiently given a (stronger) $\mathsf{ERM}_{\mathsf{DIS}}$ oracle for the class DIS of disagreements between $c, c' \in C$. We emphasize that URejectron can also be used for multi-class learning as it does not use training labels, and can be paired with any classifier trained separately. This advantage of URejectron over Rejectron comes at the cost of requiring a stronger base classifier to be used for ERM, and may lead to examples being unnecessarily rejected.

In Figure 1 we illustrate our algorithms for the class $C$ of halfspaces. A natural idea would be to train a halfspace to distinguish unlabeled training and test examples—intuitively, one can safely reject anything that is clearly distinguishable as test without increasing ▌$_P$. However, this on its own is insufficient. See for example group (2) of examples in Figure 1, which cannot be distinguished from training data by a halfspace. This is precisely why having test examples is absolutely necessary. Indeed, it allows us to use an ERM oracle to $C$ to PQ-learn $C$. We also present:

**Transductive analysis** A similar analysis of Rejectron in a transductive setting gives error and rejection bounds directly on the test examples. The bounds here are with respect to a stronger white-box adversary who need not even choose a test set $\tilde{\mathbf{x}}$ iid from a distribution. Such an adversary chooses the test set with knowledge of $P, f, h$ and $\mathbf{x}$. In particular, first $h$ is chosen based on $\mathbf{x}$ and $\mathbf{y}$; then the adversary chooses the test set $\tilde{\mathbf{x}}$ based on all available information; and finally, $S$ is chosen. We introduce a novel notion of *false rejection*, where we reject a test example that was in fact chosen from $P$ and not modified by an adversary. Theorem 4.3 gives bounds that are similar in spirit to Theorem 4.2 but for the harsher transductive setting.

**Agnostic bounds** Thus far, we have considered the realizable setting where the target $f \in C$. In agnostic learning (Kearns et al. [1992]), there is an arbitrary distribution $\mu$ over $X \times \{0, 1\}$ and the goal is to learn a classifier that is nearly as accurate as the best classifier in $C$. In our setting, we assume that there is a known $\eta \geq 0$ such that the train and test distributions $\mu$ and $\tilde{\mu}$ over $X \times \{0, 1\}$ satisfy that there is some function $f \in C$ that has error at most $\eta$ with respect to both $\mu$ and $\tilde{\mu}$. Unfortunately, we show that in such a setting one cannot guarantee less than $\Omega(\sqrt{\eta})$ errors and rejections, but we show that Rejectron nearly achieves such guarantees.

**Experiments** As a proof of concept, we perform simple controlled experiments on the task of handwritten letter classification using lower-case English letters from the EMNIST dataset (Cohen et al. [2017]). In one setup, to mimic a spamming adversary, after a classifier $h$ is trained, test examples are identified on which $h$ errs and are repeated many times in the test set. Existing SC algorithms (no matter how robust) will fail on such an example since they all choose $S$ without using unlabeled test examples—as long as an adversary can find even a single erroneous example, it can simply repeat it. In the second setup, we consider a natural test distribution which consists of a mix of lower- and upper-case letters, while the training set was only lower-case letters.

## 2  Related work

The redaction model combines SC and transductive learning, which have each been extensively studied, separately. We first discuss prior work on these topics, which (with the notable exception of online SC) has generally been considered when test examples are from the same distribution as training examples.

**Selective classification** Selective classification go by various names including "classification with a reject option" and "reliable learning." To the best of our knowledge, prior work has not considered SC using unlabeled samples from $Q \neq P$. Early learning theory work by Rivest and Sloan [1988] required a guarantee of 0 test errors and few rejections. However, Kivinen [1990] showed that, for this definition, even learning rectangles under uniform distributions $P = Q$ requires exponential number of examples (as cited by Hopkins et al. [2019] which like much other work therefore makes further assumptions on $P$ and $Q$). Most of this work assumes the same training and test distributions, without adversarial modification. Kanade et al. [2009] give a SC reduction to an agnostic learner (similar in spirit to our reduction to ERM) but again for the case of $P = Q$.

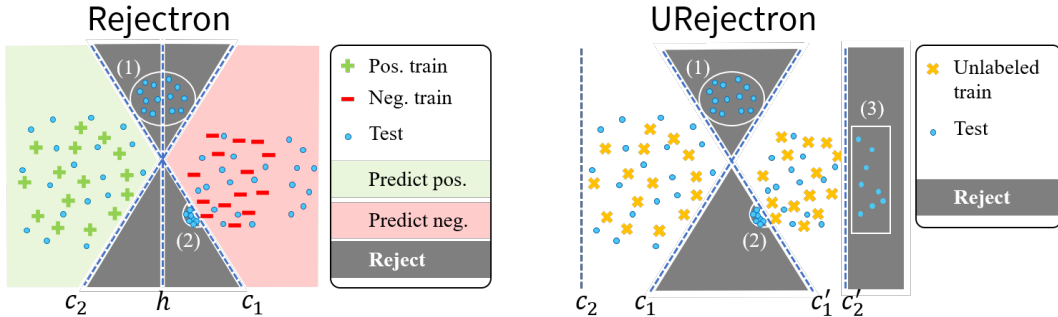

**Figure 1:** Our algorithm (and unsupervised variant) for learning $C$=halfspaces. Rejectron (left) first trains $h$ on labeled training data, then finds other candidate classifiers $c_1, c_2$, such that $h$ and $c_i$ have high disagreement on $\tilde{x}$ and low disagreement on $x$, and rejects examples where $h$ and $c_i$ disagree. URejectron (right) aims to distinguish *unlabeled train* and test examples using pairs of classifiers $c_i, c_i'$ that agree on training data but disagree on many tests. Both reject: (1) clearly unpredictable examples which are very far from train and (2) a suspiciously dense cluster of tests which might all be positive despite being close to negatives. URejectron also rejects (3).

A notable exception is the work in *online* SC, where an *arbitrary sequence* of examples is presented one-by-one with immediate error feedback. This work includes the "knows-what-it-knows" algorithm [Li et al., 2011], and Sayedi et al. [2010] exhibit an interesting trade-off between the number of mistakes and the number of rejections in such settings. However, basic classes such as intervals on the line are impossible to learn in these harsh online formulations. Interestingly, our division into labeled train and unlabeled test seems to make the problem easier than in the harsh online model.

We now discuss related work which considers $Q \neq P$, but where classifiers must predict everywhere without the possibility of outputting ▊.

**Robustness to Adversarial Examples** There is ongoing effort to devise methods for learning predictors that are robust to adversarial examples [Szegedy et al., 2013, Biggio et al., 2013, Goodfellow et al., 2015] at test time. Such work typically assumes that the adversarial examples are perturbations of honest examples chosen from $P$. The main objective is to learn a classifier that has high robust accuracy, meaning that with high probability, the classifier will answer correctly even if the test point was an adversarially perturbed example. Empirical work has mainly focused on training deep learning based classifiers to be more robust [e.g., Madry et al., 2018, Wong and Kolter, 2018, Zhang et al., 2019]. Kang et al. [2019] consider the fact that perturbations may not be known in advance, and some work [e.g., Pang et al., 2018] addresses the problem of identifying adversarial examples. We emphasize that as opposed to this line of work, we consider *arbitrary* test examples and use SC.

Detecting adversarial examples has been studied in practice, but Carlini and Wagner [2017] study ten proposed heuristics and are able to bypass all of them. Our algorithms also require a sufficiently large set of unlabeled test examples. The use of unlabeled data for improving robustness has also been empirically explored recently [e.g., Carmon et al., 2019, Stanforth et al., 2019, Zhai et al., 2019].

**Covariate Shift** The literature on learning with covariate shift is too large to survey here, see, e.g., the book by Quionero-Candela et al. [2009] and the references therein. To achieve guarantees, it is often assumed that the support of $Q$ is contained in the support of $P$. Like our work, many of these approaches use unlabeled data from $Q$ [e.g., Huang et al., 2007, Ben-David and Urner, 2012]. Ben-David and Urner [2012] show that learning with covariate-shift is intractable, in the worst case, without such assumptions. In this work we overcome this negative result, and obtain guarantees for arbitrary $Q$, using SC. In summary, prior work on covariate shift that guarantees low test/target error requires strong assumptions regarding the distributions. This motivates our model of covariate shift with rejections.

# 3 Learning with redaction

Henceforth, we assume a fixed class $C$ of $c : X \to Y$ from domain $X$ to $Y = \{0, 1\}$,[3] and let $d$ be the VC dimension of $C$. We now describe the two settings for SC. We use the same algorithm in both settings, so it can be viewed as two justifications for the same algorithm. The PQ model provides guarantees with respect to future examples from the test distribution, while the transductive model provides guarantees with respect to arbitrary test examples chosen by an all-powerful adversary. Interestingly, the transductive analysis is somewhat simpler and is used in the PQ analysis.

## 3.1 PQ learning

In the *PQ* setting, an SC learner $h|_S = L(\mathbf{x}, f(\mathbf{x}), \tilde{\mathbf{x}})$ is given $n$ labeled examples $\mathbf{x} = (x_1, \dots, x_n)$ drawn iid $\mathbf{x} \sim P^n$, labels $f(\mathbf{x}) = (f(x_1), \dots, f(x_n))$ for some unknown $f \in C$, and $n$ unlabeled examples $\tilde{\mathbf{x}} \sim Q^n$. $L$ outputs $h : X \to Y$ and $S \subseteq X$. The adversary (or nature) chooses $Q$ based only on $f, P$ and knowledge of the learning algorithm $L$. The definition of PQ learning is given in Definition 1.1. Performance is measured in terms of $\mathrm{err}_Q$ on future examples from $Q$ and $\blacksquare_P$ (rather than the more obvious $\blacksquare_Q$). Rejection rates on $P$ (and $Q$) can be estimated from held out data, if so desired. The quantities $\blacksquare_P, \blacksquare_Q$ can be related and a small $\blacksquare_P$ implies few rejections on future examples from $Q$ wherever it "overlaps" with $P$ by which we mean $Q(x) \leq \Lambda \cdot P(x)$ for some constant $\Lambda$. In particular, for any $\Lambda \geq 0, S \subseteq X$,

$$\Pr_{x \sim Q}\left[x \notin S \text{ and } Q(x) \leq \Lambda P(x)\right] \leq \Lambda \, \blacksquare_P(S). \tag{3}$$

Lemma G.1 proves the above, in addition to $\blacksquare_Q \leq \blacksquare_P + |P - Q|_{\mathrm{TV}}$. But this can be quite loose and a tighter bound is given in Appendix G. If $\blacksquare_P = 0$ then all $x \sim Q$ that lie in $P$'s support would necessarily be classified (i.e., $x \in S$).

It is also worth mentioning that a PQ-learner can also be used to guarantee $\mathrm{err}_P + \blacksquare_P \leq \epsilon$ meaning that it has *accuracy* $\Pr_P[h|_S(x) = f(x)] \geq 1 - \epsilon$ with respect to $P$ (like a normal PAC learner) but is also simultaneously robust to $Q$. Claim H.1 shows this and an additional property that PQ learners can be made robust with respect to any polynomial number of different $Q$'s.

## 3.2 Transductive setting with white-box adversary

In the *transductive* setting, there is no $Q$ and instead empirical analogs $\mathrm{err}_{\mathbf{x}}$ and $\blacksquare_{\mathbf{x}}$ of error and rejection rates are defined as follows, for arbitrary $\mathbf{x} \in X^n$:

$$\mathrm{err}_{\mathbf{x}}(h|_S, f) := \frac{1}{n}|\{i \in [n] : f(x_i) \neq h(x_i) \text{ and } x_i \in S\}| \tag{4}$$

$$\blacksquare_{\mathbf{x}}(S) := \frac{1}{n}\left|\{i \in [n] : x_i \notin S\}\right| \tag{5}$$

Again, $h, f$ and $S$ may be omitted when clear from context.

In this setting, the learner first chooses $h$ using only $\mathbf{x} \sim P^n$ and $f(\mathbf{x})$. Then, a *true* test set $\mathbf{z} \sim P^n$ is drawn. Based on all available information $(\mathbf{x}, \mathbf{z}, f, h,$ and the code for learner $L)$ the adversary modifies any number of examples from $\mathbf{z}$ to create *arbitrary* test set $\tilde{\mathbf{x}} \in X^n$. Finally, the learner chooses $S$ based on $\mathbf{x}, f(\mathbf{x})$, and $\tilde{\mathbf{x}}$. Performance is measured in terms of $\mathrm{err}_{\tilde{\mathbf{x}}} + \blacksquare_{\mathbf{z}}$ rather than $\mathrm{err}_Q + \blacksquare_P$, because $\mathbf{z} \sim P^n$. One can bound $\blacksquare_{\tilde{\mathbf{x}}}$ in terms of $\blacksquare_{\mathbf{z}}$ for any $\mathbf{z}, \tilde{\mathbf{x}} \in X^n$ and $S \subseteq X$, as follows:

$$\blacksquare_{\tilde{\mathbf{x}}} \leq \blacksquare_{\mathbf{z}} + \Delta(\mathbf{z}, \tilde{\mathbf{x}}), \quad \text{where} \quad \Delta(\mathbf{z}, \tilde{\mathbf{x}}) := \frac{1}{n}\left|\{i \in [n] : z_i \neq \tilde{x}_i\}\right|. \tag{6}$$

The hamming distance $\Delta(\mathbf{z}, \tilde{\mathbf{x}})$ is the transductive analog of $|P - Q|_{\mathrm{TV}}$. The following bounds the "false rejections," those unmodified examples that are rejected:

$$\frac{1}{n}\left|\{i \in [n] : \tilde{x}_i \notin S \text{ and } \tilde{x}_i = z_i\}\right| \leq \blacksquare_{\mathbf{z}}(S). \tag{7}$$

Both eqs. (6) and (7) follow by definition of $\blacksquare_{(\cdot)}$.

Rejectron(train $\mathbf{x} \in X^n$, labels $\mathbf{y} \in Y^n$, test $\tilde{\mathbf{x}} \in X^n$, error $\epsilon \in [0, 1]$, weight $\Lambda = n + 1$) :

- $h := \mathsf{ERM}(\mathbf{x}, \mathbf{y})$ # assume black box oracle ERM to minimize errors
- For $t = 1, 2, 3, \ldots$ :
    1. $S_t := \{x \in X : h(x) = c_1(x) = \ldots = c_{t-1}(x)\}$ # So $S_1 = X$
    2. Choose $c_t \in C$ to maximize $s_t(c) := \mathrm{err}_{\tilde{\mathbf{x}}}(h|_{S_t}, c) - \Lambda \cdot \mathrm{err}_{\mathbf{x}}(h, c)$ over $c \in C$
       # Lemma 4.1 shows how to maximize $s_t$ using ERM (err is defined in eq. (4))
    3. If $s_t(c_t) \leq \epsilon$, then stop and return $h|_{S_t}$

**Figure 2:** The Rejectron algorithm takes labeled training examples and unlabeled test examples as input, and it outputs a selective classifier $h|_S$ that predicts $h(x)$ for $x \in S$ (and rejects all $x \notin S$). Parameter $\epsilon$ controls the trade-off between errors and rejections and can be set to $\epsilon = \tilde{\Theta}(\sqrt{d/n})$ to balance the two. The weight $\Lambda$ parameter is set to its default value of $n + 1$ for realizable (noiseless) learning but should be lower for agnostic learning.

**White-box adversaries** The all-powerful transductive adversary is sometimes called "white box" in the sense that it can choose its examples while looking "inside" $h$ rather than using $h$ as a black box. While it cannot choose $\tilde{\mathbf{x}}$ with knowledge of $S$, it know what $S$ will be as a function of $\tilde{\mathbf{x}}$ if the learner is deterministic, as our algorithms are. Also, we note that the generalization analysis may be extended to a white-box model where the adversary chooses $Q$ knowing $h$, but it is cumbersome even to denote probabilities over $\tilde{\mathbf{x}} \sim Q^n$ when $Q$ itself can depend on $\mathbf{x} \sim P^n$.

## 4 Algorithms and guarantees

We assume that we have a deterministic oracle $\mathsf{ERM} = \mathsf{ERM}_C$ which, given a set of labeled examples from $X \times Y$, outputs a classifier $c \in C$ of minimal error. Figure 2 describes our algorithm Rejectron. It takes as input a set of labeled training examples $(\mathbf{x}, \mathbf{y})$, where $\mathbf{x} \in X^n$ and $\mathbf{y} \in Y^n$, and a set of test examples $\tilde{\mathbf{x}} \in X^n$ along with an error parameter $\epsilon > 0$ that trades off errors and rejections. A value for $\epsilon$ that theoretically balances these is in Theorems 4.2 and 4.3.

**Lemma 4.1** (Computational efficiency). *For any* $\mathbf{x}, \tilde{\mathbf{x}} \in X^n$, $\mathbf{y} \in Y^n$, $\epsilon > 0$ *and* $\Lambda \in \mathbb{N}$, Rejectron$(\mathbf{x}, \mathbf{y}, \tilde{\mathbf{x}}, \epsilon, \Lambda)$ *outputs* $S_{T+1}$ *for* $T \leq \lfloor 1/\epsilon \rfloor$. *Further, each iteration can be implemented using one call to* $\mathsf{ERM}$ *on at most* $(\Lambda + 1)n$ *examples and* $O(n)$ *evaluations of classifiers in* $C$.

The intuition behind this Lemma is as follows. To implement a single step of the algorithm, one runs an ERM oracle on an artificial dataset consisting of: $\Lambda$ copies of each training example $(x_i, h(x_i))$ and 1 copy of each test example $(\tilde{x}_i, 1 - h(\tilde{x}_i))$, where the training examples are labeled by $h$ and the test examples have the opposite labels that would have been assigned by $h$. It is not hard to see that the error rate on this dataset is linearly related to $s_t$ (defined in Step 2 in Rejectron) hence the ERM oracle indirectly maximizes $s_t$. To bound the number of iterations $T < 1/\epsilon$, note that during every iteration Rejectron abstains on $> \epsilon$ additional fraction of $\tilde{\mathbf{x}}$. Proof is provided in Appendix H.

Note that since we assume ERM is deterministic, the Rejectron algorithm is also deterministic. This efficient reduction to ERM, together with the following imply that Rejectron is a PQ learner:

**Theorem 4.2** (PQ guarantees). *For any* $n \in \mathbb{N}$, $\delta > 0$, $f \in C$ *and distributions* $P, Q$ *over* $X$:

$$\Pr_{\mathbf{x} \sim P^n, \tilde{\mathbf{x}} \sim Q^n}[\mathrm{err}_Q \leq 2\epsilon^* \wedge \blacksquare_P \leq \epsilon^*] \geq 1 - \delta,$$

*where* $\epsilon^* = \sqrt{\frac{8d \ln 2n}{n}} + \frac{8 \ln 16/\delta}{n}$ *and* $h|_S = \mathsf{Rejectron}(\mathbf{x}, f(\mathbf{x}), \tilde{\mathbf{x}}, \epsilon^*)$.

More generally, Theorem A.5 shows that, by varying parameter $\epsilon$, one can achieve any trade-off between $\mathrm{err}_Q \leq O(\epsilon)$ and $\blacksquare_P \leq \tilde{O}(\frac{d}{n\epsilon})$. The analogous transductive guarantee is:

**Theorem 4.3** (Transductive). *For any* $n \in \mathbb{N}$, $\delta > 0$, $f \in C$ *and dist. $P$ over $X$:*

$$\Pr_{\mathbf{x}, \mathbf{z} \sim P^n}\left[\forall \tilde{\mathbf{x}} \in X^n : \mathrm{err}_{\tilde{\mathbf{x}}}(h|_S) \leq \epsilon^* \wedge \blacksquare_{\mathbf{z}}(S) \leq \epsilon^*\right] \geq 1 - \delta,$$

*where* $\epsilon^* = \sqrt{\frac{2d}{n} \log 2n} + \frac{1}{n} \log \frac{1}{\delta}$ *and* $h|_S = \mathsf{Rejectron}(\mathbf{x}, f(\mathbf{x}), \tilde{\mathbf{x}}, \epsilon^*)$.

---

URejectron(train $\mathbf{x} \in X^n$, test $\tilde{\mathbf{x}} \in X^n$, error $\epsilon \in [0, 1]$, weight $\Lambda = n + 1$) :

- For $t = 1, 2, 3, \ldots$ :
    1. $S_t := \{x \in X : c_1(x) = c'_1(x) \wedge \cdots \wedge c_{t-1}(x) = c'_{t-1}(x)\}$  # So $S_1 = X$
    2. Choose $c_t, c'_t \in C$ to maximize $s_t(c, c') := \text{err}_{\tilde{\mathbf{x}}}(c'|_{S_t}, c) - \Lambda \cdot \text{err}_{\mathbf{x}}(c', c)$
        # Lemma B.1 shows how to maximize $s_t$ using $\text{ERM}_{\text{DIS}}$ (DIS is defined in eq. (8))
    3. If $s_t(c_t, c'_t) \leq \epsilon$, then stop and return $S_t$

---

**Figure 3:** The URejectron unsupervised algorithm takes unlabeled training examples and unlabeled test examples as input, and it outputs a set $S \subseteq X$ where classification should take place.

One thinks of $\mathbf{z}$ as the real test examples and $\tilde{\mathbf{x}}$ as an arbitrary adversarial modification, not necessarily iid. Equation (7) means that this implies $\leq \epsilon^*$ errors on unmodified examples. As discussed earlier, the guarantee above holds for *any* $\tilde{\mathbf{x}}$ chosen by a white-box adversary, which may depend on $\mathbf{x}$ and $f$, and thus on $h$ (since $h = \text{ERM}(\mathbf{x}, f(\mathbf{x}))$ is determined by $\mathbf{x}$ and $f$). More generally, Theorem A.2 shows that, by varying parameter $\epsilon$, one can trade-off $\text{err}_{\tilde{\mathbf{x}}} \leq \epsilon$ and $\blacksquare_{\mathbf{z}} \leq \tilde{O}(\frac{d}{n\epsilon})$.

We note that Theorems 4.2 and 4.3 generalize in a rather straightforward manner to the case in which an adversary can inject additional training examples to form $\mathbf{x}' \supseteq \mathbf{x}$ which contains $\mathbf{x}$. We give the proof sketch of Theorem 4.3, since it is slightly simpler than Theorem 4.2. Full proofs are in Appendix A.

*Proof sketch for Theorem 4.3.* To show $\text{err}_{\tilde{\mathbf{x}}} \leq \epsilon^*$, fix any $f, \mathbf{x}, \tilde{\mathbf{x}}$. Since $h = \text{ERM}(\mathbf{x}, f(\mathbf{x}))$ and $f \in C$, this implies that $h$ has zero training error, i.e., $\text{err}_{\mathbf{x}}(h, f) = 0$. Hence $s_t(f) = \text{err}_{\tilde{\mathbf{x}}}(h|_{S_t}, f)$ and the algorithm cannot terminate with $\text{err}_{\tilde{\mathbf{x}}}(h|_{S_t}, f) > \epsilon$ since it could have selected $c_t = f$.

To prove $\blacksquare_{\mathbf{z}} \leq \epsilon^*$, observe that Rejectron never rejects any training $\mathbf{x}$. This follows from the fact that $\Lambda > n$, together with the fact that $h(x_i) = f(x_i)$ for every $i \in [n]$ which follows, in turn, from the facts that $f \in C$ and $h = \text{ERM}(\mathbf{x}, f(\mathbf{x}))$. Now $\mathbf{x}$ and $\mathbf{z}$ are identically distributed. By a generalization-like bound (Lemma A.1), with probability $\geq 1 - \delta$ there is no classifier for which selects all of $\mathbf{x}$ and yet rejects with probability greater than $\epsilon^*$ on $\mathbf{z}$ for $T \leq 1/\epsilon^*$ (by Lemma 4.1). $\square$

Unfortunately, the above bounds are worse than standard $\tilde{O}(d/n)$ VC-bounds for $P = Q$, but the following lower-bound shows that $\tilde{O}(\sqrt{d/n})$ is tight for some class $C$.

**Theorem 4.4** (PQ lower bound). *There exists a constant $K > 0$ such that: for any $d \geq 1$, there is a concept class $C$ of VC dimension $d$, distributions $P$ and $Q$, such that for any $n \geq 2d$ and learner $L : X^n \times Y^n \times X^n \to Y^X \times 2^X$, there exists $f \in C$ with*

$$\mathbb{E}_{\substack{\mathbf{x} \sim P^n \\ \tilde{\mathbf{x}} \sim Q^n}} \left[ \blacksquare_P + \text{err}_Q \right] \geq K \sqrt{\frac{d}{n}}, \quad \text{where} \quad h|_S = L(\mathbf{x}, f(\mathbf{x}), \tilde{\mathbf{x}}).$$

Note that since $P$ and $Q$ are fixed, independent of the learner $L$, the unlabeled test examples from $Q$ are not useful for the learner as they could simulate as many samples from $Q$ as they would like on their own. Thus, the lower bound holds even given $n$ training examples and $m$ unlabeled test examples, for arbitrarily large $m$.

Theorem 4.4 implies that the learner needs at least $n = \Omega(d/\epsilon^2)$ labeled training examples to get the $\epsilon$ error plus rejection guarantee. However, it leaves open the possibility that many fewer than $m = \tilde{O}(d/\epsilon^2)$ test examples are needed. Theorem F.1 is a lower bound in the transductive case which shows that both $m, n$ must be at least $\Omega(d/\epsilon^2)$. This $\Omega(\sqrt{d/\min\{m, n\}})$ lower bound implies that one needs both $\Omega(d/\epsilon^2)$ training and test examples to guarantee $\epsilon$ error plus rejections. This is partly why, for simplicity, aside from the Theorem F.1, our analysis takes $m = n$. The proofs of these two lower bounds are in Appendix F.

**Unsupervised selection algorithm.** Our unsupervised selection algorithm URejectron is described in Figure 3. It takes as input only train and test examples $\mathbf{x}, \tilde{\mathbf{x}} \in X^n$ along with an error parameter $\epsilon$

recommended to be $\tilde{\Theta}(\sqrt{d/n})$, and it outputs a set $S$ of the selected elements. URejectron requires a more powerful black-box ERM—we show that URejectron can be implemented efficiently if one can perform ERM with respect to the family of binary classifiers that are disagreements (xors) between two classifiers. For classifiers $c, c' : X \to Y$, define $\text{dis}_{c,c'} : X \to \{0,1\}$ and DIS as follows:

$$\text{dis}_{c,c'}(x) := \begin{cases} 1 & \text{if } c(x) \neq c'(x) \\ 0 & \text{otherwise} \end{cases} \quad \text{and} \quad \text{DIS} := \{\text{dis}_{c,c'} : c, c' \in C\}. \tag{8}$$

Lemma B.1 shows how URejectron is implemented efficiently with an $\text{ERM}_{\text{DIS}}$ oracle. Also, we show nearly identical guarantees to those of Theorem 4.3 for URejectron:

**Theorem 4.5** (Unsupervised). *For any $n \in \mathbb{N}$, any $\delta \geq 0$, and any distribution $P$ over $X$:*

$$\Pr_{\mathbf{x},\mathbf{z} \sim P^n} \left[ \forall f \in C, \tilde{\mathbf{x}} \in X^n : \left( \text{err}_{\tilde{\mathbf{x}}}(h|_S) \leq \epsilon^* \right) \wedge \left( \mathbb{I}_{\mathbf{z}}(S) \leq \epsilon^* \right) \right] \geq 1 - \delta,$$

*where $\epsilon^* = \sqrt{\frac{2d}{n} \log 2n} + \frac{1}{n} \log \frac{1}{\delta}$, $S = \text{URejectron}(\mathbf{x}, \tilde{\mathbf{x}}, \epsilon^*)$ and $h = \text{ERM}_C(\mathbf{x}, f(\mathbf{x}))$.*

The proof is given in Appendix B and follows from Theorem B.2 which shows that by varying parameter $\epsilon$, one can achieve any trade-off $\text{err}_{\tilde{\mathbf{x}}} \leq \epsilon$ and $\mathbb{I}_{\mathbf{z}} \leq \tilde{O}(\frac{d}{n\epsilon})$. Since one runs URejectron without labels, it has guarantees with respect to any empirical risk minimizer $h$ which may be chosen separately, and its output is also suitable for a multi-class problem.

**Massart noise.** We also consider two non-realizable models. First, we consider the Massart noise model, where there is an arbitrary (possibly adversarial) noise rate $\eta(x) \leq \eta$ chosen for each example. We show that Rejectron is a PQ learner in the Massart noise model with $\eta < 1/2$, assuming an ERM oracle and an additional $N = \tilde{O}\left(\frac{dn^2}{\delta^2(1-2\eta)^2}\right)$ examples from $P$. See Appendix C for details.

**A semi-agnostic setting.** We also consider the following semi-agnostic model. For an arbitrary distribution $D$ over $X \times Y$, again with $Y = \{0,1\}$, the analogous notions of rejection and error are:

$$\mathbb{I}_D(S) := \Pr_{(x,y) \sim D}[x \notin S] \quad \text{and} \quad \text{err}_D(h|_S) := \Pr_{(x,y) \sim D}[h(x) \neq y \wedge x \in S]$$

In standard agnostic learning with respect to $D$, we suppose there is some classifier $f \in C$ with error $\text{err}_D(f) \leq \eta$ and we aim to find a classifier whose generalization error is not much greater than $\eta$. In that setting, one can of course choose $\eta_{\text{opt}} := \min_{f \in C} \text{err}_D(f)$. For well-fitting models, where there is some classifier with very low error, $\eta$ may be small.

To prove any guarantees in our setting, the test distribution must somehow be related to the training distribution. To tie together the respective training and test distributions $\mu, \tilde{\mu}$ over $X \times Y$, we suppose we know $\eta$ such that both $\text{err}_\mu(f) \leq \eta$ and $\text{err}_{\tilde{\mu}}(f) \leq \eta$ for some $f \in C$. Even with these conditions, Lemma D.1 shows that one cannot simultaneously guarantee error rate on $\tilde{\mu}$ and rejection rate on $\mu$ less than $\sqrt{\eta/8}$, and Theorem D.2 shows that our Rejectron algorithm achieves a similar upper bound. This suggests that PQ-learning (i.e., adversarial SC) may be especially challenging in settings where ML is not able to achieve low error $\eta$.

# 5 Conclusions

The fundamental theorem of statistical learning states that an ERM algorithm for class $C$ is asymptotically nearly optimal requiring $\tilde{\Theta}(d/n)$ labeled examples for learning arbitrary distributions when $P = Q$ [see, e.g., Shalev-Shwartz and Ben-David, 2014]. This paper can be viewed as a generalization of this theorem to the case where $P \neq Q$, obtaining $\tilde{\Theta}(\sqrt{d/n})$ rates. When $P = Q$, unlabeled samples from $Q$ are readily available by ignoring labels of some training data, but unlabeled test samples are necessary when $P \neq Q$. No prior such guarantee was known for arbitrary $P \neq Q$, even for simple classes such as intervals, perhaps because it may have seemed impossible to guarantee anything meaningful in the general case.

The practical implications are that, to address learning in the face of adversaries beyond perturbations (or drastic covariate shift), unlabeled examples and abstaining from classifying may be necessary. In this model, the learner can beat an unbounded white-box adversary. Even the simple approach of training a classifier to distinguish unlabeled train vs. test examples may be adequate in some applications, though for theoretical guarantees one requires somewhat more sophisticated algorithms.

## Broader Impact

In adversarial learning, this work can benefit users when adversarial examples are correctly identified. It can harm users by misidentifying such examples, and the misidentifications of examples as suspicious could have negative consequences just like misclassifications. This work ideally could benefit groups who are underrepresented in training data, by abstaining rather than performing harmful incorrect classification. However, it could also harm such groups: (a) by providing system designers an alternative to collecting fully representative data if possible; (b) by harmfully abstaining at different rates for different groups; (c) when those labels would have otherwise been correct but are instead being withheld; and (d) by identifying them when they would prefer to remain anonymous.

Our experiments on handwriting recognition have few ethical concerns but also have less ecological validity than real-world experiments on classifying explicit images or medical scans.

**A note of caution.** Inequities may be caused by using training data that differs from the test distribution on which the classifier is used. For instance, in classifying a person's gender from a facial image, Buolamwini and Gebru [2018] have demonstrated that commercial classifiers are highly inaccurate on dark-skinned faces, likely because they were trained on light-skinned faces. In such cases, it is preferable to collect a more diverse training sample even if it comes at greater expense, or in some cases to abstain from using machine learning altogether. In such cases, $PQ$ learning should *not* be used, as an unbalanced distribution of rejections can also be harmful.[4]

## Acknowledgments and Disclosure of Funding

We thank the anonymous reviewers for their thoughtful and helpful feedback. We do not have funding to disclose.

## Footnotes

*Author order is alphabetical.

[2]Learning with an all-negative training set (trivial in standard learning) is a useful "anomaly detection" setting in adversarial learning, e.g., when one aims to classify illegal images without any illegal examples at train time or abnormal scans not present at train time.

[3]For simplicity, the theoretical model is defined for binary classification, though our experiments illustrate a multi-class application. To avoid measure-theoretic issues, we assume $X$ is countably infinite or finite.

[4]We are grateful to an anonymous reviewer who pointed out that gender classification is an example of when *not* to use $PQ$ learning.

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
