[Supplementary Material · beyond_perturbations_supp_usletter.pdf]

# A  Rejectron analysis (realizable)

In this section, we present the analysis of Rejectron in the realizable case $f \in C$. Say a classifier $c$ is *consistent* if $c(\mathbf{x}) = f(\mathbf{x})$ makes 0 training errors. Theorem 4.3 provides transductive guarantees on the empirical error and rejection rates, while Theorem 4.2 provides generalization guarantees that apply to future examples from $P, Q$. Both of these theorems exhibit trade-offs between error and rejection rates. At a high level, their analysis has the following structure:

- Rejectron selects a consistent $h = \mathsf{ERM}(\mathbf{x}, f(\mathbf{x}))$, since we are in the realizable case.
- Each $c_t$ is a consistent classifier that disagrees with $h|_{S_t}$ on the tests $\tilde{\mathbf{x}}$ as much as possible, with $s_t(c_t) = \mathrm{err}_{\tilde{\mathbf{x}}}(h|_{S_t}, c_t)$ (since $\mathrm{err}_{\mathbf{x}}(h, c_t) = 0$). This follows the facts that $\Lambda > n$, $s_t(h) = 0$, and $s_t(c) < 0$ for any inconsistent $c$. (The algorithm is defined for general $\Lambda < n$ for the agnostic analysis later.)
- Therefore, when the algorithm terminates on iteration $T$, it has empirical test error $\mathrm{err}_{\tilde{\mathbf{x}}}(h|_{S_T}, f) \le \epsilon$ otherwise it could have chosen $c_t = f$.
- The number of iterations $T < 1/\epsilon$ since on each iteration an additional $\epsilon$ fraction of $\tilde{\mathbf{x}}$ is removed from $S_t$. Lemma 4.1 states this and shows how to use an ERM oracle on an artificial dataset to efficiently find $c_t$.
- All training examples $x_i$ are in $S$ since each $c_t$ and $h$ agree on all $x_i$.
- Transductive error and rejection bounds:
    1. For error, we have already argued that the empirical error $\mathrm{err}_{\tilde{\mathbf{x}}} \le \epsilon$.
    2. For rejection, Lemma A.1 states that it is unlikely that there would be any choice of $h, \mathbf{c} = (c_1, \dots, c_T)$ where the resulting $S(h, \mathbf{c}) := \{x \in X : h(x) = c_1(x) = \dots = c_T(x)\}$ would contain all training examples but reject (abstain on) many "true" test examples $z_i$ since $\mathbf{x}$ and $\mathbf{z}$ are identically distributed. The proof uses Sauer's lemma.
- Generalization error and rejection bounds:
    1. For error, Lemma A.3 states that it is unlikely that there is any $h, \mathbf{c}$ such that $\mathrm{err}_{\tilde{\mathbf{x}}}(h|_{S(h,\mathbf{c})}) \le \epsilon$ yet $\mathrm{err}_Q(h|_{S(h,\mathbf{c})}) > 2\epsilon$.
    2. For rejection rate, Lemma A.4 uses VC bounds to show that it is unlikely that $\blacksquare_P(S(h, \mathbf{c})) > \epsilon$ while $\blacksquare_{\mathbf{x}}(S(h, \mathbf{c})) = 0$.

    Both proofs use Sauer's lemma.

We next move to the transductive analysis since it is simpler, and it is also used as a stepping stone to the generalization analysis.

## A.1  Transductive guarantees (realizable)

Note that Rejectron rejects any $x \notin S$, where $S = S(h, \mathbf{c})$ is defined by

$$S(h, \mathbf{c}) := \{x \in X : h(x) = c_1(x) = c_2(x) = \dots = c_T(x)\}. \tag{9}$$

In what follows, we prove the transductive analogue of a "generalization" guarantee for arbitrary $h \in C, \mathbf{c} \in C^T$. This will be useful when proving Theorem 4.3.

**Lemma A.1.** *For any $T, n \in \mathbb{N}$, any $\delta \ge 0$, and $\epsilon = \frac{1}{n}\left(d(T+1)\log(2n) + \log\frac{1}{\delta}\right)$:*

$$\Pr_{\mathbf{x},\mathbf{z} \sim P^n}\left[\exists \mathbf{c} \in C^T, h \in C : (\blacksquare_{\mathbf{x}}(S(h,\mathbf{c})) = 0) \wedge (\blacksquare_{\mathbf{z}}(S(h,\mathbf{c})) > \epsilon)\right] \le \delta.$$

This lemma is proven in Appendix E. Using it, we can show a trade-off between error and rejection rate for the transductive case.

**Theorem A.2.** *For any $n \in \mathbb{N}$, any $\epsilon, \delta \ge 0$, any $f \in C$:*

$$\forall \mathbf{x}, \tilde{\mathbf{x}} \in X^n : \mathrm{err}_{\tilde{\mathbf{x}}}(h|_S, f) \le \epsilon, \tag{10}$$

*where $h|_S = \mathsf{Rejectron}(\mathbf{x}, f(\mathbf{x}), \tilde{\mathbf{x}}, \epsilon)$, and for any distribution $P$ over $X$,*

$$\Pr_{\mathbf{x},\mathbf{z} \sim P^n}\left[\forall \tilde{\mathbf{x}} \in X^n : \blacksquare_{\mathbf{z}}(S) \le \frac{1}{n}\left(\frac{2d}{\epsilon}\log(2n) + \log\frac{1}{\delta}\right)\right] \ge 1 - \delta. \tag{11}$$

We note that a natural alternative formalization of Equation (11) would be to require that

$$\Pr_{\mathbf{x} \sim P^n} \left[ \forall \tilde{\mathbf{x}} \in X^n \; : \; \mathbb{1}_P(S) \leq \frac{1}{n} \left( \frac{2d}{\epsilon} \log(2n) + \log \frac{1}{\delta} \right) \right] \geq 1 - \delta.$$

However, the formalization of Equation (11) is stronger, as it guarantees that the rejection probability is small, even if the adversary is *"white-box"* and chooses $\tilde{\mathbf{x}}$ after seeing $\mathbf{z}$.

*Proof of Theorem A.2.* We start by proving eq. (10). To this end, fix any $n \in \mathbb{N}$, any $\epsilon > 0$, any $f \in C$, and any $\mathbf{x}, \tilde{\mathbf{x}} \in X^n$. Let $h = \mathsf{ERM}(\mathbf{x}, f(\mathbf{x}))$. Since we are in the realizable case, this implies that $h$ has zero training error, i.e., $\mathrm{err}_{\mathbf{x}}(h, f) = 0$, and hence $s_t(h) = \mathrm{err}_{\tilde{\mathbf{x}}}(h|_{S_t}, f)$ for all $t$. Thus, the algorithm cannot terminate on any iteration where $\mathrm{err}_{\tilde{\mathbf{x}}}(h|_{S_t}, f) > \epsilon$ since it can always select $c_t = f \in C$. This proves Equation (10).

It remains to prove eq. (11). By Lemma 4.1, $T = \lfloor 1/\epsilon \rfloor$ is an upper bound on the number of completed iterations of the algorithm. WLOG there are exactly $T$ iterations because if there were actually $T' < T$ iterations, simply "pad" them with $c_{T'+1} = \ldots = c_T = h$ which doesn't change $S$.

We note that the algorithm selects all training examples. This follows from the fact that $\Lambda > n$, together with the fact that $h(x_i) = f(x_i)$ for every $i \in [n]$, where the latter follows from the fact that $f \in C$ and $h = \mathsf{ERM}(\mathbf{x}, f(\mathbf{x}))$. By Lemma A.1, with probability $\geq 1 - \delta$ there are no choices $h \in C, \mathbf{c} = (c_1, \ldots, c_T) \in C^T$ for which $S(h, \mathbf{c})$ contains all $x_i$'s but is missing $\geq \epsilon'$ fraction of $\mathbf{z}$ for $\epsilon' = \frac{1}{n} \left( \frac{2d}{\epsilon} \log(2n) + \log \frac{1}{\delta} \right)$ since $T + 1 \leq 2/\epsilon$. $\qquad\square$

Theorem 4.3 is a trivial corollary of Theorem A.2.

*Proof of Theorem 4.3.* Recall $\epsilon^* = \sqrt{\frac{2d}{n} \log 2n} + \frac{1}{n} \log \frac{1}{\delta}$ and $h|_S = \mathsf{Rejectron}(\mathbf{x}, f(\mathbf{x}), \tilde{\mathbf{x}}, \epsilon^*)$. The proof follows from Theorem A.2 and the fact that:

$$\frac{1}{n} \left( \frac{2d}{\epsilon^*} \log 2n + \log \frac{1}{\delta} \right) \leq \frac{2d \log 2n}{n\sqrt{\frac{2d}{n} \log 2n}} + \frac{1}{n} \log \frac{1}{\delta} = \epsilon^*.$$

$\qquad\square$

## A.2 Generalization guarantees (realizable)

Before we state our generalization guarantees, analogous to Lemma A.1 above, we prove that low test error and low training rejection rates imply, with high probability, low generalization error and rejection rates.

**Lemma A.3.** *For any $\delta > 0, \epsilon \geq \frac{8 \ln 8/\delta}{n} + \sqrt{\frac{8d \ln 2n}{n}}, T \leq 1/\epsilon$, any $f, h \in C$ and any distribution $Q$ over $X$,*

$$\Pr_{\mathbf{z} \sim Q^n} \left[ \exists \mathbf{c} \in C^T \; : \; \left( \mathrm{err}_Q(h|_{S(h,\mathbf{c})}, f) > 2\epsilon \right) \wedge \left( \mathrm{err}_{\mathbf{z}}(h|_{S(h,\mathbf{c})}, f) \leq \epsilon \right) \right] \leq \delta.$$

**Lemma A.4.** *For any $T \geq 1$, any $f \in C$ and any distribution $P$ over $X$,*

$$\Pr_{\mathbf{x} \sim P^n} \left[ \exists h \in C, \mathbf{c} \in C^T \; : \; \left( \mathbb{1}_P(S(h, \mathbf{c})) > \xi \right) \wedge \left( \mathbb{1}_{\mathbf{x}}(S(h, \mathbf{c})) = 0 \right) \right] \leq \delta,$$

*where $\xi = \frac{2}{n}(d(T+1) \log(2n) + \log \frac{2}{\delta})$. Also,*

$$\Pr_{\mathbf{x} \sim P^n} \left[ \exists h \in C, \mathbf{c} \in C^T \; : \; \left( \mathbb{1}_P(S(h, \mathbf{c})) > 2\alpha \right) \wedge \left( \mathbb{1}_{\mathbf{x}}(S(h, \mathbf{c})) \leq \alpha \right) \right] \leq \delta,$$

*for any $\alpha \geq \frac{8}{n}(d(T+1) \ln(2n) + \ln \frac{8}{\delta})$.*

We mention that the first inequality in Lemma A.4 is used to provide generalization guarantees in the realizable setting, whereas the latter inequality is used to provide guarantees in the semi-agnostic setting.

**Theorem A.5.** *For any $n \in \mathbb{N}$ and $\delta > 0$, any $\epsilon \geq \sqrt{\frac{8d \ln 2n}{n}} + \frac{8 \ln 8/\delta}{n}$, any $f \in C$ and any distributions $P, Q$ over $X$:*

$$\forall \mathbf{x} \in X^n : \Pr_{\tilde{\mathbf{x}} \sim Q^n} \left[ \mathrm{err}_Q(h|_S) \leq 2\epsilon \right] \geq 1 - \delta, \tag{12}$$

*where $h|_S := \mathsf{Rejectron}(\mathbf{x}, f(\mathbf{x}), \tilde{\mathbf{x}}, \epsilon)$. Furthermore, for any $\epsilon \geq 0$,*

$$\Pr_{\mathbf{x} \sim P^n} \left[ \forall \tilde{\mathbf{x}} \in X^n : \blacksquare_P \leq \frac{2}{n} \left( \frac{2d}{\epsilon} \log 2n + \log \frac{2}{\delta} \right) \right] \geq 1 - \delta. \tag{13}$$

*Proof of Theorem A.5.* Let $T = \lfloor 1/\epsilon \rfloor$ be an upper bound on the number of iterations. We first prove eq. (12). Since the ERM algorithm is assumed to be deterministic, the function $h$ is uniquely determined by $\mathbf{x}$ and $f$. By Theorem A.2 (Equation (10)), the set $S$ has the property that $\mathrm{err}_{\tilde{\mathbf{x}}}(h|_S) \leq \epsilon$ (with certainty) for all $\mathbf{x}, \tilde{\mathbf{x}}$. By Lemma A.3, with probability at most $\delta$ there exists a choice of $h, \mathbf{c}$ which would lead to $\mathrm{err}_Q(h|_S) > 2\epsilon$ and $\mathrm{err}_{\tilde{\mathbf{x}}}(h|_S) \leq \epsilon$, implying eq. (12).

For eq. (13), as we argued in the proof of Theorem A.2, the fact that $\Lambda > n$, together with the fact we are in the realizable case (i.e., $\mathbf{y} = f(\mathbf{x})$), implies that we select all training examples. Because of this and the fact that $T + 1 \leq 2/\epsilon$, Lemma A.4 implies eq. (13). $\qquad\square$

Theorem 4.2 is a trivial corollary of Theorem A.5.

*Proof of Theorem 4.2.* Recall that $\epsilon^* = \sqrt{\frac{8d \ln 2n}{n}} + \frac{8 \ln 16/\delta}{n}$.

Equation (12) implies that $\Pr[\mathrm{err}_Q \leq 2\epsilon^*] \geq 1 - \delta/2$ and eq. (13) implies,

$$\Pr_{\mathbf{x} \sim P^n} \left[ \forall \mathbf{z} \in X^n : \blacksquare_P \leq \frac{2}{n} \left( \frac{2d}{\epsilon^*} \log 2n + \log \frac{4}{\delta} \right) \right] \geq 1 - \frac{\delta}{2}.$$

Further, note that $\log_2 r \leq 2 \ln r$ for $r \geq 1$ and hence, using $\epsilon^* > \sqrt{\frac{8d \ln 2n}{n}}$,

$$\frac{2}{n} \left( \frac{2d}{\epsilon^*} \log 2n + \log \frac{4}{\delta} \right) \leq \frac{8d}{n\epsilon^*} \ln 2n + \frac{4}{n} \ln \frac{4}{\delta} < \sqrt{\frac{8d \ln 2n}{n}} + \frac{4}{n} \ln \frac{4}{\delta} \leq \epsilon^*.$$

The proof is completed by the union bound. $\qquad\square$

# B   Analysis of Urejectron

In this section we present a transductive analysis of $\mathsf{URejectron}$, again in the realizable case. We begin with its computational efficiency.

**Lemma B.1** ($\mathsf{URejectron}$ computational efficiency). *For any $\mathbf{x}, \tilde{\mathbf{x}} \in X^n, \epsilon > 0$ and $\Lambda \in \mathbb{N}$, $\mathsf{URejectron}$ outputs $S_{T+1}$ for $T \leq \lfloor 1/\epsilon \rfloor$. Further, each iteration can be implemented using one call to $\mathsf{ERM}_{\mathsf{DIS}}$, as defined in eq. (8), on at most $(\Lambda + 1)n$ examples and $O(n)$ evaluations of classifiers in $C$.*

The proof of this lemma is nearly identical to that of Lemma 4.1.

*Proof of Lemma B.1.* The argument that $T \leq \lfloor 1/\epsilon \rfloor$ follows for the same reason as before, replacing eq. (45) with:

$$|\{i : x_i \in S_t\}| - |\{i : x_i \in S_{t+1}\}| = |\{i : x_i \in S_t \wedge c_t(x_i) \neq c_t'(x_i)\}| = n \, \mathrm{err}_{\tilde{\mathbf{x}}}(c_t|_{S_t}, c_t') \geq n\epsilon.$$

For efficiency, again all that needs to be stored are the subset of indices $Z_t = \{i \mid \tilde{x}_i \in S_t\}$ and the classifiers $c_1, c_1', \dots, c_T, c_T'$ necessary to compute $S$. To implement iteration $t$ using the $\mathsf{ERM}_{\mathsf{DIS}}$ oracle, construct a dataset consisting of each training example, labeled by 0, repeated $\Lambda$ times, and each test example in $\tilde{x}_i \in S_t$, labeled 1, included just once. The accuracy of $\mathrm{dis}_{c,c'}$ on this dataset is easily seen to differ by a constant from $s_t(c, c')$, hence running $\mathsf{ERM}_{\mathsf{DIS}}$ maximizes $s_t$. $\qquad\square$

The following Theorem exhibits the trade-off between accuracy and rejections.

**Theorem B.2.** *For any $n \in \mathbb{N}$, any $\epsilon \geq 0$,*

$$\forall \mathbf{x}, \tilde{\mathbf{x}} \in X^n, f \in C \: : \: \mathrm{err}_{\tilde{\mathbf{x}}}(h|_S) \leq \epsilon, \tag{14}$$

*where $S = \mathsf{URejectron}(\mathbf{x}, \tilde{\mathbf{x}}, \epsilon)$ and $h = \mathsf{ERM}_C(\mathbf{x}, f(\mathbf{x}))$. Furthermore, for any $\delta > 0$ and any distribution $P$ over $X$:*

$$\Pr_{\mathbf{x}, \mathbf{z} \sim P^n}\left[ \blacksquare_{\mathbf{z}}(S) \leq \frac{1}{n}\left( \frac{2d \log 2n}{\epsilon} + \log 1/\delta \right) \right] \geq 1 - \delta. \tag{15}$$

Before we prove Theorem B.2 we provide some generalization bounds that will be used in the proof. To this end, given a family $G$ of classifiers $g : X \to \{0, 1\}$, following Blumer et al. [1989], define:

$$\Pi_G[2n] := \max_{\mathbf{w} \in X^{2n}} |\{g(\mathbf{w}) : g \in G\}|. \tag{16}$$

**Lemma B.3** (Transductive train-test bounds). *For any $n \in \mathbb{N}$, any distribution $P$ over a domain $X$, any set $G$ of classifiers over $X$, and any $\epsilon > 0$,*

$$\Pr_{\mathbf{x}, \mathbf{z} \sim P^n}\left[ \exists g \in G \: : \: \left( \frac{1}{n} \sum_i g(z_i) \geq \epsilon \right) \wedge \left( \frac{1}{n}\sum_i g(x_i) = 0 \right) \right] \leq \Pi_G[2n] 2^{-\epsilon n} \tag{17}$$

*and*

$$\Pr_{\mathbf{x}, \mathbf{z} \sim P^n}\left[ \exists g \in G \: : \: \frac{1}{n}\sum_i g(z_i) \geq \frac{1+\alpha}{n}\sum_i g(x_i) + \epsilon \right] \leq \Pi_G[2n] e^{-\frac{2\alpha}{(2+\alpha)^2}\epsilon n}. \tag{18}$$

The proof of this lemma is deferred to Appendix E. (Note eq. (18) is used for the agnostic analysis later.)

*Proof of Theorem B.2.* We denote for $T \geq 1$ and classifier vectors $\mathbf{c}, \mathbf{c}' \in C^T$:

$$\delta_{\mathbf{c}, \mathbf{c}'}(x) := \max_{i \in [T]} \mathrm{dis}_{c_i, c_i'}(x) = \begin{cases} 1 & \text{if } c_i(x) \neq c_i'(x) \text{ for some } i \in [T] \\ 0 & \text{otherwise.} \end{cases}$$

$$\Delta_T := \left\{ \delta_{\mathbf{c}, \mathbf{c}'} \: : \: \mathbf{c}, \mathbf{c}' \in C^T \right\}.$$

Thus the output of $\mathsf{URejectron}$ is $S_{T+1} = \{x \in X \: : \: \delta_{\mathbf{c}, \mathbf{c}'}(x) = 0\}$ for the vectors $\mathbf{c} = (c_1, \ldots, c_T)$ and $\mathbf{c}' = (c_1', \ldots, c_T')$ chosen by the algorithm.

Let $T$ be the final iteration of the algorithm so that the output of the algorithm is $S = S_{T+1}$. Note that $\mathrm{err}_{\mathbf{x}}(f, h) = 0$, by definition of $\mathsf{ERM}_C$, so $s_{T+1}(f, h) = \mathrm{err}_{\tilde{\mathbf{x}}}(h|_S) \leq \epsilon$ (otherwise the algorithm would have chosen $c = h, c' = f$ instead of halting) which implies eq. (14).

By Lemma B.1, WLOG we can take $T = \lfloor 1/\epsilon \rfloor$ by padding with classifiers $c_t = c_t'$.

We next claim that $x_i \notin S_t$ for all $i \in [n]$, i.e., $\delta_{\mathbf{c}, \mathbf{c}'}(x_i) = 0$. This is because the algorithm is run with $\Lambda = n + 1$, so any disagreement $c_t(x_i) \neq c_t'(x_i)$ would result in a negative score $s_t(c_t, c_t')$. (But a zero score is always possible by choosing $c_t = c_t'$.) Thus we must have the property that $\mathrm{dis}_{c_t', c_t}(x_i) = 0$ and hence $\delta_{\mathbf{c}, \mathbf{c}'}(x_i) = 0$. Now, it is not difficult to see that $\Pi_{\Delta_T}[2n] \leq (2n)^{2d/\epsilon}$ because, by Sauer's lemma, there are at most $N = (2n)^d$ different labelings of $2n$ examples by classifiers from $C$, hence there are at most $\binom{N}{2}^T \leq (2n)^{2dT}$ disagreement labelings for $T \leq 1/\epsilon$ pairs. Thus for $\xi = \frac{1}{n}\left( \frac{2d \log 2n}{\epsilon} + \log 1/\delta \right)$, by Lemma B.3,

$$\Pr_{\mathbf{x}, \mathbf{z} \sim P^n}\left[ \forall g \in \Delta_T \text{ s.t. } \sum_i g(x_i) = 0 \: : \: \frac{1}{n}\sum_i g(z_i) \leq \xi \right] \geq 1 - \Pi_{\Delta_T}[2n] 2^{-\xi n} \geq 1 - \delta.$$

If this $1 - \delta$ likely event happens, then also $\blacksquare_{\mathbf{z}}(S) = \frac{1}{n}\sum_i \delta_{\mathbf{c}, \mathbf{c}'}(z_i) \leq \xi$ for the algorithm choices $\mathbf{c}, \mathbf{c}'$. $\qquad\square$

*Proof of Theorem 4.5.* The proof follows from Theorem B.2 and the fact that,

$$\frac{1}{n}\left( \frac{2d \log 2n}{\epsilon^*} + \log 1/\delta \right) \leq \frac{2d \log 2n}{n\sqrt{\frac{2d \log 2n}{n}}} + \frac{\log 1/\delta}{n} = \epsilon^*.$$

$\qquad\square$

## C    Massart Noise

This section shows that we can PQ learn in the Massart noise model. The Massart model [Massart et al., 2006] is defined with respect to a noise rate $\eta < 1/2$ and function (abusing notation) $\eta : X \to [0, \eta]$:

**Definition C.1** (Massart Noise Model). *Let $P$ be a distribution on $X$, $\eta < 1/2$, and $0 \leq \eta(x) \leq \eta$ for all $x \in X$. The Massart distribution $P_{\eta,f}$ with respect to $f$ over $(x, y) \in X \times Y$ is defined as follows: first $x \sim P$ is chosen and then $y = f(x)$ with probability $1 - \eta(x)$ and $y = 1 - f(x)$ with probability $\eta(x)$.*

When clear from context, we omit $f$ and write $P_\eta = P_{\eta,f}$. The following lemma relates the *clean* error rate $\mathrm{err}_P(h, f) = \Pr_P[h(x) \neq f(x)]$ and *noisy* error rate $\mathrm{err}_{P_\eta} = \Pr_{(x,y) \sim P_\eta}[h(x) \neq y]$. Later, we will show how to drive the clean error arbitrarily close to 0 using an ERM.

**Lemma C.2.** *For any classifier $g : X \to Y$, any $\eta < 1/2$, $f \in C$, and any distribution $P_\eta$ corrupted with Massart noise:*

$$(1 - 2\eta) \, \mathrm{err}_P(g) \leq \mathrm{err}_{P_\eta}(g) - \mathrm{OPT},$$

*where $\mathrm{OPT} = \min_{h \in C} \mathrm{err}_{P_\eta}(h) = \mathbb{E}_{x \sim P}[\eta(x)]$.*

*Proof.* By definition of the noisy error rate of $g$ under $P_\eta$, observe the following:

$$
\begin{aligned}
\mathrm{err}_{P_\eta}(g) &= \Pr_{(x,y) \sim P_\eta} [g(x) \neq y] \\
&= \mathbb{E}_{x \sim P} [\eta(x)\mathbf{1}\{g(x) = f(x)\} + (1 - \eta(x))\mathbf{1}\{g(x) \neq f(x)\}] \\
&= \mathbb{E}_{x \sim P} [\eta(x)(1 - \mathbf{1}\{g(x) \neq f(x)\}) + (1 - \eta(x))\mathbf{1}\{g(x) \neq f(x)\}] \\
&= \mathbb{E}_{x \sim P} [\eta(x)] + \mathbb{E}_{x \sim P} [(1 - 2\eta(x))\mathbf{1}\{g(x) \neq f(x)\}] \\
&= \mathrm{OPT} + \mathbb{E}_{x \sim P} [(1 - 2\eta(x))\mathbf{1}\{g(x) \neq f(x)\}] \\
&\geq \mathrm{OPT} + (1 - 2\eta) \, \mathbb{E}_{x \sim P}[\mathbf{1}\{g(x) \neq f(x)\}] \\
&= \mathrm{OPT} + (1 - 2\eta) \, \mathrm{err}_P(g),
\end{aligned}
$$

where the last inequality follows from the fact that $\eta(x) \leq \eta$ for every $x \in X$. Rearranging the terms concludes the proof. $\qquad\square$

The following lemma shows that using an extra $N = \tilde{O}\left(\frac{dn^2}{\delta^2(1-2\eta)^2}\right)$ i.i.d. examples $(\mathbf{x}', \mathbf{y}') \sim P_\eta^N$, we can "denoise" the $n$ held-out examples $(\mathbf{x}, \mathbf{y}) \sim P_\eta^n$ with $\hat{h} = \mathrm{ERM}_C(\mathbf{x}', \mathbf{y}')$, and then run Rejectron on $(\mathbf{x}, \hat{h}(\mathbf{x}))$. This shows that we can PQ learn $C$ under Massart noise.

**Lemma C.3** (Massart denoising). *For any $f \in C$ and any distribution $P$ over $X$, any $\eta < 1/2$ and $\eta : X \to [0, \eta]$, let $P_\eta$ be the corresponding Massart distribution over $(x, y)$. For any $n \in \mathbb{N}$, let $(\mathbf{x}, \mathbf{y}) = (x_1, y_1), \dots, (x_n, y_n) \sim P_\eta$ be i.i.d. examples sampled from $P_\eta$. Then,*

$$\Pr_{(\mathbf{x}', \mathbf{y}') \sim P_\eta^N} \left[\mathrm{err}_{\mathbf{x}}(\hat{h}, f) = 0\right] \geq 1 - \delta,$$

*where $\hat{h} = \mathrm{ERM}_C(\mathbf{x}', \mathbf{y}')$ and $N = O\left(\frac{dn^2 + \log(2/\delta)}{\delta^2(1-2\eta)^2}\right)$.*

*Proof.* By agnostic learning guarantees for $\mathrm{ERM}_C$, we have that for any $\epsilon', \delta > 0$:

$$\Pr_{(\mathbf{x}', \mathbf{y}') \sim P_\eta^N} \left[\mathrm{err}_{P_\eta}(\hat{h}) \leq \mathrm{OPT} + \epsilon'\right] \geq 1 - \frac{\delta}{2},$$

where $\hat{h} = \mathrm{ERM}_C(\mathbf{x}', \mathbf{y}')$ and $N = O(\frac{d + \log(2/\delta)}{\epsilon'^2})$. By Lemma C.2, choosing $\epsilon' = \frac{\delta}{2n}(1 - 2\eta)$ guarantees that the clean error rate $\mathrm{err}_P(\hat{h}) \leq \frac{\delta}{2n}$. Since, $(\mathbf{x}, \mathbf{y}) \sim P_\eta^n$ are independent held-out examples, by a union bound, we get that $\mathrm{err}_{\mathbf{x}}(\hat{h}, f) = 0$ with probability $1 - \delta$. $\qquad\square$

This yields an easy algorithm and corollary: simply use the $N$ examples $\mathbf{x}', \mathbf{y}'$ to denoise the $n$ labels for $\mathbf{x}$ and then run the Rejectron algorithm.

**Corollary C.4** (PQ guarantees under Massart noise). *For any $n \in \mathbb{N}, \delta > 0, f \in C$ and distributions $P, Q$ over $X$, any $\eta < 1/2$ and $\eta : X \to [0, \eta]$, let $P_\eta$ be the corresponding Massart distribution over $(x, y)$. Then,*

$$\Pr_{(\mathbf{x}',\mathbf{y}')\sim P_\eta^N,(\mathbf{x},\mathbf{y})\sim P_\eta^n,\tilde{\mathbf{x}}\sim Q^n}[\mathrm{err}_Q \leq 2\epsilon^* \wedge \|_P \leq \epsilon^*] \geq 1 - \delta,$$

*where $\epsilon^* = \sqrt{8\frac{d\ln 2n}{n}} + \frac{8\ln 32/\delta}{n}$, $N = O\left(\frac{dn^2 + \log(2/\delta)}{\delta^2(1-2\eta)^2}\right)$, $\hat{h} = \mathrm{ERM}_C(\mathbf{x}', \mathbf{y}')$, and $h|_S = $* Rejectron$(\mathbf{x}, \hat{h}(\mathbf{x}), \tilde{\mathbf{x}}, \epsilon^*)$.

# D  Semi-agnostic analysis

In agnostic learning, the learner is given pairs $(x, y)$ from some unknown distribution $\mu$, and it is assumed that there exists some (unknown) $f \in C, \eta \geq 0$ such that

$$\mathrm{err}_\mu(f) := \Pr_{(x,y)\sim\mu}[y \neq f(x)] \leq \eta.$$

In this work, we consider the case where the test distribution $\tilde{\mu}$ may be (arbitrarily) different from the train distribution $\mu$, yet we require the existence of parameters $\eta, \tilde{\eta} \geq 0$ and an (unknown) $f \in C$ such that

$$\mathrm{err}_\mu(f) \leq \eta \quad \text{and} \quad \mathrm{err}_{\tilde{\mu}}(f) \leq \tilde{\eta}.^{[5]}$$

Moreover, in this work we assume that $\eta$ and $\tilde{\eta}$ are known. Unfortunately, even with this additional assumption, agnostic learning is challenging when $\mu \neq \tilde{\mu}$ and one cannot achieve guarantees near $\max\{\eta, \tilde{\eta}\}$ as one would hope, as we demonstrate below.

In what follows, we slightly abuse notation and write $(\mathbf{x}, \mathbf{y}) \sim D^n$ to denote $(x_i, y_i)$ drawn iid from $D$ for $i = 1, 2, \ldots, n$. The definitions of error and rejection with respect to such a distribution are:

$$\|_D(S) := \Pr_{(x,y)\sim D}[x \notin S]$$

$$\mathrm{err}_D(h|_S) := \Pr_{(x,y)\sim D}[h(x) \neq y \wedge x \in S]$$

We prove the following lower bound.

**Lemma D.1.** *There exists a family of binary classifiers $C$ of VC dimension 1, such that for any $\eta, \tilde{\eta} \in [0, 1/2]$ and $n \geq 1$, and for any selective classification algorithm $L : X^n \times Y^n \times X^n \to Y^X \times 2^X$ there exists $\mu, \tilde{\mu}$ over $X \times Y$ and $f \in C$ such that:*

$$\mathbb{E}_{\substack{(\mathbf{x},\mathbf{y})\sim\mu^n \\ (\tilde{\mathbf{x}},\tilde{\mathbf{y}})\sim\tilde{\mu}^n}}[\mathrm{err}_{\tilde{\mu}}(h|_S) + \|_\mu(S)] \geq \max\left\{\sqrt{\eta/8}, \tilde{\eta}\right\}.$$

*where $h|_S = L(\mathbf{x}, \mathbf{y}, \mathbf{z})$ and where $\mathrm{err}_\mu(f) \leq \eta$ and $\mathrm{err}_{\tilde{\mu}}(f) \leq \tilde{\eta}$.*

The proof is deferred to Section F.

We now show that Rejectron can be used to achieve nearly this guarantee. Recall that in the realizable setting, we fixed $\Lambda = n + 1$ in Rejectron. In this semi-agnostic setting, we will set $\Lambda$ as a function of $\eta$, hence our learner requires knowledge of $\eta$ unlike standard agnostic learning when $\mu = \tilde{\mu}$.

**Theorem D.2** (Agnostic generalization). *For any $n \in \mathbb{N}$, any $\delta, \gamma \in (0, 1)$, any $\eta, \tilde{\eta} \in [0, 1)$, and any distributions $\mu, \tilde{\mu}$ over $X \times Y$ such that that $\mathrm{err}_\mu(f) \leq \eta$ and $\mathrm{err}_{\tilde{\mu}}(f) \leq \tilde{\eta}$ simultaneously for some $f \in C$:*

$$\Pr_{\substack{(\mathbf{x},\mathbf{y})\sim\mu^n \\ (\tilde{\mathbf{x}},\tilde{\mathbf{y}})\sim\tilde{\mu}^n}}\left[\left(\mathrm{err}_{\tilde{\mu}}(h|_S) \leq 2\sqrt{2\eta} + 2\tilde{\eta} + 4\epsilon^*\right) \wedge \left(\|_\mu(S) \leq 4\sqrt{2\eta} + 4\epsilon^*\right)\right] \geq 1 - \delta,$$

*where $\epsilon^* = 4\sqrt{\frac{d\ln 2n + \ln 48/\delta}{n}}$, $\Lambda^* = \sqrt{\frac{1}{8\eta + (\epsilon^*)^2}}$, and $h|_S = $ Rejectron$(\mathbf{x}, \mathbf{y}, \tilde{\mathbf{x}}, \epsilon^*, \Lambda^*)$.*

A few points of comparison are worth making:

- When $\eta = \tilde{\eta} = 0$, one recovers guarantees that are slightly worse than those in the realizable (see Theorem 4.2).

- In standard agnostic learning, where $\mu$ and $\tilde{\mu}$ are identical, and thus $\eta = \tilde{\eta}$, one can set $S = X$ (i.e., select everything) and ERM guarantees $\mathrm{err}\left(h|_S(\tilde{\mathbf{x}}), \tilde{\mathbf{y}}\right) \leq \eta + \epsilon$ w.h.p. for $n$ sufficiently large.

- The above theorem can be used to bound $\blacksquare_{\tilde{\mu}}$ using the following lemma:

**Lemma D.3.** *For any $S \subseteq X$, $f, h \in Y^X$ and distributions $\mu, \tilde{\mu}$ over $X \times Y$:*

$$\blacksquare_{\tilde{\mu}}(S) \leq \blacksquare_{\mu}(S) + |\mu_X - \tilde{\mu}_X|_{\mathsf{TV}} \leq \blacksquare_{\mu}(S) + |\mu - \tilde{\mu}|_{\mathsf{TV}},$$

*where $\mu_X, \tilde{\mu}_X$ are the marginal distributions of $\mu, \tilde{\mu}$ over $X$.*

*Proof.* The lemma follows from Lemma G.1 applied to $P = \mu_X, Q = \tilde{\mu}_X$, and from the fact that the total variation between marginal distributions is no greater than the originals: $|\mu_X - \tilde{\mu}_X|_{\mathsf{TV}} \leq |\mu - \tilde{\mu}|_{\mathsf{TV}}$. □

As before, it will be useful (and easier) to first analyze the transductive case. In this case, it will be useful to further abuse notation and define, for any $\mathbf{y}, \mathbf{y}' \in \{0, 1, \blacksquare\}^n$,

$$\mathrm{err}(\mathbf{y}, \mathbf{y}') := \frac{1}{n}\left|\{i \,:\, y_i = 1 - y_i'\}\right|.$$

Using this, we will show:

**Theorem D.4** (Agnostic transductive). *For any $n \in \mathbb{N}$, $\epsilon, \delta, \Lambda \geq 0$, $f \in C$:*

$$\forall \mathbf{x}, \tilde{\mathbf{x}} \in X^n, \mathbf{y}, \tilde{\mathbf{y}} \in Y^n \,:\, \mathrm{err}(h|_S(\tilde{\mathbf{x}}), \tilde{\mathbf{y}}) \leq \epsilon + 2\Lambda \cdot \mathrm{err}(f(\mathbf{x}), \mathbf{y}) + \mathrm{err}(f(\tilde{\mathbf{x}}), \tilde{\mathbf{y}}), \qquad (19)$$

*where $h|_S = \mathsf{Rejectron}(\mathbf{x}, \mathbf{y}, \tilde{\mathbf{x}}, \epsilon, \Lambda)$. Furthermore,*

$$\Pr_{\mathbf{x}, \mathbf{z} \sim P^n}\left[\forall \mathbf{y} \in Y^n, \tilde{\mathbf{x}} \in X^n \,:\, \blacksquare_{\mathbf{z}}(S) \leq 2\Lambda^{-1} + \frac{9}{n}\left(\frac{d \ln 2n}{\epsilon} + \frac{\ln 1/\delta}{2}\right)\right] \geq 1 - \delta. \qquad (20)$$

The above bounds suggest the natural choice of $\Lambda = \eta^{-1/2}$ if $\mathrm{err}(f(\mathbf{x}), \mathbf{y}) \approx \eta$. The following two Lemmas will be used in its proof.

**Lemma D.5.** *For any $n \in \mathbb{N}$, $\epsilon, \Lambda \geq 0$, $\mathbf{x}, \mathbf{z} \in X^n, \mathbf{y} \in Y^n$: $\blacksquare_{\mathbf{x}}(S) \leq 1/\Lambda$ where $h|_S = \mathsf{Rejectron}(\mathbf{x}, \mathbf{y}, \mathbf{z}, \epsilon, \Lambda)$.*

*Proof.* Note that for each iteration $t$ of the algorithm $\mathsf{Rejectron}(\mathbf{x}, \mathbf{y}, \mathbf{z}, \epsilon, \Lambda)$,

$$\sum_{i \in [n]: z_i \in S_t} |c_t(z_i) - h(z_i)| - \Lambda \sum_{i \in [n]} |c_t(x_i) - h(x_i)| \geq 0,$$

because $c_t$ maximizes the above quantity over $C$, and that quantity is zero at $c_t = h \in C$. Also note that $x \notin S$ if and only if $|c_t(x) - h(x)| = 1$ for some $t$. More specifically, for each $i \in [n]$ such that $z_i \notin S$ there exists a *unique* $t \in [T]$ such that $z_i \in S_t$, and yet $z_i \notin S_{t+1}$, where the latter occurs when $|c_t(z_i) - h(z_i)| = 1$. Thus the total number of test and train rejections can be related as follows:

$$n \geq n\blacksquare_{\mathbf{z}}(S) = \sum_{t \in [T]}\sum_{i \in [n]: z_i \in S_t} |c_t(z_i) - h(z_i)| \geq \sum_{t \in [T]}\Lambda\sum_{i \in [n]} |c_t(x_i) - h(x_i)| \geq n\Lambda\blacksquare_{\mathbf{x}}(S).$$

Dividing both sides by $n \cdot \Lambda$ gives the lemma. □

The following lemma is proven in Appendix E.

**Lemma D.6.** *For any $T, n \in \mathbb{N}$, any $\delta \geq 0$, and $\epsilon = \frac{9}{2n}\left(d(T+1)\ln(2n) + \ln\frac{1}{\delta}\right)$:*

$$\Pr_{\mathbf{x}, \mathbf{z} \sim P^n}\left[\exists \mathbf{c} \in C^T, h \in C \,:\, \blacksquare_{\mathbf{z}}(S(h, \mathbf{c})) > 2\blacksquare_{\mathbf{x}}(S(h, \mathbf{c})) + \epsilon\right] \leq \delta.$$

Using these, we can now prove the transductive agnostic theorem.

*Proof of Theorem D.4.* To prove Equation (19), first fix any $\mathbf{x}, \tilde{\mathbf{x}} \in X^n, \mathbf{y}, \tilde{\mathbf{y}} \in Y^n, f \in C$. Since $f \in C$ the output $h = \mathsf{ERM}_C(\mathbf{x}, \mathbf{y})$ satisfies $\mathrm{err}(h(\mathbf{x}), \mathbf{y}) \leq \mathrm{err}(f(\mathbf{x}), \mathbf{y})$. By the triangle inequality, this implies that

$$\mathrm{err}_{\mathbf{x}}(h, f) = \frac{1}{n} \sum_{i \in [n]} |h(x_i) - f(x_i)| \leq \frac{1}{n} \sum_{i \in [n]} \left( |h(x_i) - y_i| + |y_i - f(x_i)| \right) \leq 2 \, \mathrm{err}(f(\mathbf{x}), \mathbf{y}). \quad (21)$$

Now suppose the algorithm Rejectron terminates on iteration $T+1$ and the output is $h|_S$ for $S = S_{T+1}$. Then by definition, for every $c \in C$,

$$s_{T+1}(c) = \mathrm{err}_{\tilde{\mathbf{x}}}(h|_S, c) - \Lambda \cdot \mathrm{err}_{\mathbf{x}}(h, c) \leq \epsilon,$$

For $c = f$ in particular,

$$\mathrm{err}_{\tilde{\mathbf{x}}}(h|_S, f) \leq \Lambda \cdot \mathrm{err}_{\mathbf{x}}(h, f) + \epsilon \leq 2\Lambda \cdot \mathrm{err}(f(\mathbf{x}), \mathbf{y}) + \epsilon.$$

Equation (19) follows from the above and the fact that

$$\mathrm{err}(h|_S(\tilde{\mathbf{x}}), \tilde{\mathbf{y}}) \leq \mathrm{err}(h|_{S_T}(\tilde{\mathbf{x}}), f(\tilde{\mathbf{x}})) + \mathrm{err}(f(\tilde{\mathbf{x}}), \tilde{\mathbf{y}}).$$

We next prove eq. (20). By Lemma D.5, $\blacksquare_{\mathbf{x}}(S) \leq 1/\Lambda$ with certainty. So by Lemma D.6 applied to the marginal distribution $P = \mu_X$ over $X$,

$$\Pr_{\mathbf{x}, \mathbf{z} \sim P^n} \left[ \exists h \in C, \mathbf{c} \in C^T : \blacksquare_{\mathbf{z}}(S(h, \mathbf{c})) > 2 \, \blacksquare_{\mathbf{x}}(S(h, \mathbf{c})) + \xi \right] \leq \delta,$$

for $\xi = \frac{9}{2n} \left( \frac{2d}{\epsilon} \ln(2n) + \ln \frac{1}{\delta} \right)$ using $T + 1 \leq 2/\epsilon$. This implies eq. (20). $\quad\square$

Returning to the generalization (distributional) agnostic case, the following theorem shows the trade-off between error and rejections:

**Theorem D.7.** *For any $n \in \mathbb{N}$ and $\delta, \Lambda \geq 0$, any $\epsilon \geq 4\sqrt{\frac{d \ln 2n + \ln 24/\delta}{n}}$, any $f \in C$, and any distributions $\mu, \tilde{\mu}$ over $X \times Y$:*

$$\Pr_{\substack{(\mathbf{x}, \mathbf{y}) \sim \mu^n \\ (\tilde{\mathbf{x}}, \tilde{\mathbf{y}}) \sim \tilde{\mu}^n}} \left[ \mathrm{err}_{\tilde{\mu}}(h|_S) \leq 8\Lambda \, \mathrm{err}_{\mu}(f) + 2 \, \mathrm{err}_{\tilde{\mu}}(f) + \Lambda \epsilon^2 + 3\epsilon \right] \geq 1 - \delta, \quad (22)$$

*where $h|_S = \mathsf{Rejectron}(\mathbf{x}, \mathbf{y}, \tilde{\mathbf{x}}, \epsilon, \Lambda)$. Furthermore,*

$$\Pr_{(\mathbf{x}, \mathbf{y}) \sim \mu^n} \left[ \forall \tilde{\mathbf{x}} \in X^n : \blacksquare_{\mu}(S) \leq \frac{2}{\Lambda} + 2\epsilon \right] \geq 1 - \delta. \quad (23)$$

The proof of this theorem will use the following lemma.

**Lemma D.8.** *For any $h \in C$, distribution $\mu$ over $X \times Y$, and $\epsilon = \frac{16}{n} \left( dT \ln 2n + \ln \frac{8}{\delta} \right)$,*

$$\Pr_{(\mathbf{x}, \mathbf{y}) \sim \mu^n} \left[ \forall \mathbf{c} \in C^T : \mathrm{err}_{\mu}(h|_{S(h, \mathbf{c})}) \leq \max \left\{ 2 \, \mathrm{err}(h|_{S(h, \mathbf{c})}(\mathbf{x}), \mathbf{y}), \epsilon \right\} \right] \geq 1 - \delta.$$

The proof of this lemma is deferred to Appendix E.

*Proof of Theorem D.7.* The proof structure follows the proof of Theorem A.5. Fix $f$. We start by proving Equation (22). Let $\zeta = \frac{16}{n} \left( \frac{2d}{\epsilon} \ln 2n + \ln \frac{24}{\delta} \right)$. By Lemma D.8,

$$\forall h \in C \quad \Pr_{(\tilde{\mathbf{x}}, \tilde{\mathbf{y}}) \sim \tilde{\mu}^n} \left[ \forall \mathbf{c} \in C^T : \mathrm{err}_{\tilde{\mu}}(h|_{S(h, \mathbf{c})}) \leq \max \left\{ 2 \, \mathrm{err}(h|_{S(h, \mathbf{c})}(\tilde{\mathbf{x}}), \tilde{\mathbf{y}}), \zeta \right\} \right] \geq 1 - \delta/3.$$

Equation (19) from Theorem D.4 states that,

$$\forall \mathbf{x}, \tilde{\mathbf{x}} \in X^n, \mathbf{y}, \tilde{\mathbf{y}} \in Y^n : \mathrm{err}(h|_S(\tilde{\mathbf{x}}), \tilde{\mathbf{y}}) \leq 2\Lambda \cdot \mathrm{err}(f(\mathbf{x}), \mathbf{y}) + \mathrm{err}(f(\tilde{\mathbf{x}}), \tilde{\mathbf{y}}) + \epsilon,$$

with certainty. We next bound $\mathrm{err}(f(\mathbf{x}),\mathbf{y})$ and $\mathrm{err}(f(\tilde{\mathbf{x}}),\tilde{\mathbf{y}})$.

Since $\epsilon^2/4 \geq \frac{4}{n}\ln\frac{3}{\delta}$, multiplicative Chernoff bounds imply that,

$$\Pr_{(\mathbf{x},\mathbf{y})\sim\mu^n}\left[\mathrm{err}(f(\mathbf{x}),\mathbf{y}) \leq 2\,\mathrm{err}_\mu(f) + \frac{\epsilon^2}{4}\right] \geq 1 - \delta/3.$$

Also, since $\epsilon/2 \geq \sqrt{\ln(3/\delta)/(2n)}$, additive Chernoff bounds imply that,

$$\Pr_{(\tilde{\mathbf{x}},\tilde{\mathbf{y}})\sim\tilde{\mu}^n}\left[\mathrm{err}(f(\tilde{\mathbf{x}}),\tilde{\mathbf{y}}) \leq \mathrm{err}_{\tilde{\mu}}(f) + \frac{\epsilon}{2}\right] \geq 1 - \delta/3$$

Combining previous four displayed inequalities, which by the union bound all hold with probability $\geq 1 - \delta$, gives,

$$\Pr_{\substack{(\mathbf{x},\mathbf{y})\sim\mu^n \\ (\tilde{\mathbf{x}},\tilde{\mathbf{y}})\sim\tilde{\mu}^n}}\left[\mathrm{err}_{\tilde{\mu}}(h|_S) \leq \max\left\{2\left(2\Lambda(2\,\mathrm{err}_\mu(f) + \epsilon^2/4) + (\mathrm{err}_{\tilde{\mu}}(f) + \epsilon/2) + \epsilon\right), \zeta\right\}\right] \geq 1 - \delta.$$

Simplifying:

$$\Pr_{\substack{(\mathbf{x},\mathbf{y})\sim\mu^n \\ (\tilde{\mathbf{x}},\tilde{\mathbf{y}})\sim\tilde{\mu}^n}}\left[\mathrm{err}_{\tilde{\mu}}(h|_S) \leq \max\left\{8\Lambda\,\mathrm{err}_\mu(f) + \Lambda\epsilon^2 + 2\,\mathrm{err}_{\tilde{\mu}}(f) + 3\epsilon, \zeta\right\}\right] \geq 1 - \delta. \qquad (24)$$

Next, we note that for our requirement of $\epsilon \geq 4\sqrt{\frac{d\ln 2n + \ln 24/\delta}{n}}$, $\zeta \leq 2\epsilon$ because:

$$\zeta = \frac{16}{n}\left(\frac{2d}{\epsilon}\ln 2n + \ln\frac{24}{\delta}\right) \leq \frac{32}{n\epsilon}\left(d\ln 2n + \ln\frac{24}{\delta}\right) \leq 2\frac{\epsilon^2}{\epsilon}.$$

Thus we can remove the maximum from eq. (24),

$$\Pr_{\substack{(\mathbf{x},\mathbf{y})\sim\mu^n \\ (\tilde{\mathbf{x}},\tilde{\mathbf{y}})\sim\tilde{\mu}^n}}\left[\mathrm{err}_{\tilde{\mu}}(h|_S) \leq 8\Lambda\,\mathrm{err}_\mu(f) + \Lambda\epsilon^2 + 2\,\mathrm{err}_{\tilde{\mu}}(f) + 3\epsilon\right] \geq 1 - \delta,$$

which is equivalent to what needed to prove in eq. (22).

We next prove eq. (23). By Lemma D.5, $\|_\mathbf{x}(S) \leq 1/\Lambda$ with certainty. So by Lemma E.2 (Equation (32)) with $\gamma = 1/2$,

$$\Pr_{\mathbf{x},\mathbf{z}\sim P^n}\left[\exists h \in C, \mathbf{c} \in C^T \;:\; \|_\mu\left(S(h,\mathbf{c})\right) > 2\,\|_\mathbf{x}\left(S(h,\mathbf{c})\right) + \xi\right] \leq \delta,$$

for $\xi = \frac{16}{n}\left(\frac{2d}{\epsilon}\ln(2n) + \ln\frac{8}{\delta}\right)$ using $T + 1 \leq 2/\epsilon$. This implies eq. (23) using the fact that,

$$\xi = \frac{16}{n}\left(\frac{2d}{\epsilon}\ln(2n) + \ln\frac{16}{\delta}\right) \leq 2 \cdot \frac{16}{n\epsilon}\left(d\ln(2n) + \ln\frac{16}{\delta}\right) \leq 2 \cdot \frac{\epsilon^2}{\epsilon} = 2\epsilon.$$

$\square$

From this theorem, our main agnostic upper-bound follows in a straightforward fashion.

*Proof of Theorem D.2.* Note that for our choice of $\Lambda^* = \sqrt{\frac{1}{8\eta + (\epsilon^*)^2}}$,

$$\begin{aligned}
\left(8\Lambda^*\,\mathrm{err}_\mu(f) + 2\,\mathrm{err}_{\tilde{\mu}}(f)\right) + \Lambda^*(\epsilon^*)^2 + 3\epsilon^* &\leq \Lambda^*(8\eta + (\epsilon^*)^2) + 2\tilde{\eta} + 3\epsilon^* \\
&= \sqrt{8\eta + (\epsilon^*)^2} + 2\tilde{\eta} + 3\epsilon^* \\
&\leq 2\sqrt{2\eta} + \epsilon^* + 2\tilde{\eta} + 3\epsilon^*,
\end{aligned}$$

using the fact that $\sqrt{a + b} \leq \sqrt{a} + \sqrt{b}$. For the chosen $\epsilon^* = 4\sqrt{\frac{d\ln 2n + \ln 48/\delta}{n}}$, Theorem D.7 implies,

$$\Pr_{\substack{(\mathbf{x},\mathbf{y})\sim\mu^n \\ (\tilde{\mathbf{x}},\tilde{\mathbf{y}})\sim\tilde{\mu}^n}}\left[\mathrm{err}_{\tilde{\mu}}(h|_S) \leq \left(8\Lambda^*\,\mathrm{err}_\mu(f) + 2\,\mathrm{err}_{\tilde{\mu}}(f)\right) + \Lambda^*(\epsilon^*)^2 + 3\epsilon^*\right] \geq 1 - \delta/2$$

$$\Pr_{\substack{(\mathbf{x},\mathbf{y})\sim\mu^n \\ (\tilde{\mathbf{x}},\tilde{\mathbf{y}})\sim\tilde{\mu}^n}}\left[\mathrm{err}_{\tilde{\mu}}(h|_S) \leq 2\sqrt{2\eta} + 2\tilde{\eta} + 4\epsilon^*\right] \geq 1 - \delta/2 \qquad (25)$$

Also note that

$$\frac{2}{\Lambda^*} + 2\epsilon^* \leq 2\sqrt{8\eta + (\epsilon^*)^2} + 2\epsilon^* \leq 4\sqrt{2\eta} + 2\epsilon^* + 2\epsilon^* \leq 4\sqrt{2\eta} + 4\epsilon^*.$$

Theorem D.7 also implies:

$$\Pr_{(\mathbf{x},\mathbf{y})\sim\mu^n}\left[\forall \tilde{\mathbf{x}} \in X^n : \ \blacksquare_\mu(S) \leq \frac{2}{\Lambda^*} + 2\epsilon^*\right] \geq 1 - \delta/2$$

$$\Pr_{(\mathbf{x},\mathbf{y})\sim\mu^n}\left[\forall \tilde{\mathbf{x}} \in X^n : \ \blacksquare_\mu(S) \leq 4\sqrt{2\eta} + 4\epsilon^*\right] \geq 1 - \delta/2$$

$$\Pr_{(\mathbf{x},\mathbf{y})\sim\mu^n}\left[\forall \tilde{\mathbf{x}} \in X^n : \ \blacksquare_{\tilde{\mu}}(S) \leq 4\sqrt{2\eta} + 4\epsilon^* + |\mu - \tilde{\mu}|_{\mathsf{TV}}\right] \geq 1 - \delta/2, \tag{26}$$

where we have used Lemma D.3 in the last step. The union bound over eq. (25) and eq. (26) proves the corollary. $\qquad\square$

# E   Generalization Lemmas

In this section we state auxiliary lemmas that relate the empirical error and rejection rates to generalization error and rejection rates.

To bound generalization, it will be useful to note that the classifiers $h|_S$ output by our algorithm are not too complex. To do this, for any $k \in \mathbb{N}$ and any classifiers $c_1, c_2, \dots, c_k : X \to Y$, define the *disagreement* function that is 1 if any of two disagree on $x$:

$$\mathsf{dis}_{c_1,\dots,c_k}(x) := \begin{cases} 0 & \text{if } c_1(x) = c_2(x) = \cdots = c_k(x) \\ 1 & \text{otherwise.} \end{cases} \tag{27}$$

Also denote by $\bar{f} = 1 - f$ and $\mathbf{c} = (c_1, \dots, c_T) \in C^T$. In these terms we can write,

$$\mathsf{dis}_{h,\mathbf{c}} = \begin{cases} 0 & \text{if } h(x) = c_1(x) = c_2(x) = \cdots = c_T(x) \\ 1 & \text{otherwise} \end{cases}$$

$$\mathsf{dis}_{\bar{f},h,\mathbf{c}} = \begin{cases} 1 & \text{if } 1 - f(x) = h(x) = c_1(x) = c_2(x) = \cdots = c_T(x) \\ 0 & \text{otherwise.} \end{cases}$$

Recall the definition of $\Pi_G[2n]$ for a family $G$ of classifiers $g : X \to \{0, 1\}$:

$$\Pi_G[2n] := \max_{\mathbf{w} \in X^{2n}} |\{g(\mathbf{w}) : g \in G\}|.$$

**Lemma E.1** (Complexity of output class). *For any $h \in C$, let*

$$\mathsf{Dis}_T := \left\{ \mathsf{dis}_{h,c_1,\dots,c_T} : h, c_1, \dots, c_T \in C \right\} \tag{28}$$

$$\mathsf{Dis}_{h,T} := \left\{ \mathsf{dis}_{h,c_1,\dots,c_T} : c_1, \dots, c_T \in C \right\}, \tag{29}$$

$$\mathsf{Dis}_{f,h,T} := \left\{ \mathsf{dis}_{f,h,c_1,\dots,c_T} : c_1, \dots, c_T \in C \right\}, \tag{30}$$

*Then $\Pi_{\mathsf{Dis}_T}[2n] \leq (2n)^{d(T+1)}$, $\Pi_{\mathsf{Dis}_{h,T}}[2n] \leq (2n)^{dT}$, and $\Pi_{\mathsf{Dis}_{f,h,T}}[2n] \leq (2n)^{dT}$, where $d$ is the VC dimension of $C$.*

*Proof.* The proof follows trivially from Sauer's lemma, since the number of labelings of $2n$ examples by any $c \in C$ is at most $(2n)^d$ and there are $T$ choices of $c_1, \dots, c_T$ and 1 choice of $h$. $\qquad\square$

**Lemma E.2** (Generalization bounds using Blumer et al. [1989]). *For any $n \in \mathbb{N}$, any distribution $P$ over a domain $X$, any set $G$ of binary classifiers over $X$, and any $\epsilon > 0$,*

$$\Pr_{\mathbf{z}\sim P^n}\left[\exists g \in G : \left(\mathop{\mathbb{E}}_{x\sim P}[g(x)] > \epsilon\right) \wedge \left(\frac{1}{n}\sum_{i\in[n]} g(z_i) = 0\right)\right] \leq 2\Pi_G[2n]2^{-\epsilon n/2}, \tag{31}$$

*and, for any $\gamma \in (0, 1)$,*

$$\Pr_{\mathbf{z} \sim P^n} \left[ \exists g \in G : \mathbb{E}_{x \sim P}[g(x)] > \max \left\{ \epsilon, \frac{1}{1-\gamma} \cdot \frac{1}{n} \sum_{i \in [n]} g(z_i) \right\} \right] \leq 8\Pi_G[2n] e^{-\frac{\gamma^2 \epsilon n}{4}}. \quad (32)$$

*Finally, for any distribution $\mu$ over $X \times Y$ and any $\gamma \in (0, 1)$,*

$$\Pr_{(\mathbf{x},\mathbf{y}) \sim \mu^n} \left[ \exists g \in G : \mathrm{err}_\mu(g) > \max \left\{ \epsilon, \frac{1}{1-\gamma} \cdot \frac{1}{n} \sum_{i \in [n]} |g(x_i) - y_i| \right\} \right] \leq 8\Pi_G[2n] e^{-\frac{\gamma^2 \epsilon n}{4}}. \quad (33)$$

*Proof.* Simply consider a binary classification problem where the target classifier is the constant 0 function, with training examples $\mathbf{z} \sim P^n$. Then the training error rate is $\sum g(z_i)/n$ and the generalization error is $\Pr_P[g(x) = 1]$. By Theorem A2.1 of Blumer et al. [1989], the probability that any $g \in G$ has 0 training error and test error greater than $\epsilon$ is at most $\Pi_G[2n]2^{-\epsilon n/2}$. Similarly eq. (32) and (33) follow from Theorem A3.1 of Blumer et al. [1989], noting that the bound holds trivially for all $g$ with $\mathbb{E}[g(x)] \leq \epsilon$. $\qquad \square$

We now prove Lemma D.8, which adapts the last bound above to the agnostic setting.

*Proof of Lemma D.8.* We would like to apply the last inequality of Lemma E.2 with $\gamma = 1/2$, but unfortunately that lemma does not apply to error rates of selective classifiers. First, consider the case where the distribution is "consistent" in that $\Pr_{x,y \sim \mu}[y = \tau(x)]$ for some arbitrary $\tau : X \rightarrow \{0, 1\}$. We can consider the modified functions,

$$g_{h,\mathbf{c}}(x) = \begin{cases} \tau(x) & \text{if } x \notin S(h, \mathbf{c}) \\ h(x) & \text{otherwise.} \end{cases}$$

It follows that $\mathrm{err}_\mu(g_{h,\mathbf{c}}) = \mathrm{err}_\mu(h|_S)$. Furthermore, the class $G = \{g_{h,\mathbf{c}} : h \in C, \mathbf{c} \in C^T\}$ satisfies $\Pi_G[2n] \leq (2n)^{dT}$ (just as we argued $\Pi_{\mathrm{Dis}_{h,T}}[2n] \leq (2n)^{dT}$) because there are $(2n)^d$ different labelings of $c$ on $2n$ elements and thus there are at most $(2n)^{dT}$ choices of $T$ of these for $\mathbf{c} \in C^T$. Thus, applying Lemma E.2 gives the lemma for consistent $\mu$.

The inconsistent case can be reduced to the consistent case by a standard trick. In particular, we will extend $X$ to $X' = X \times \{0, 1\}$ by appending a latent (hidden) copy of $y$, call it $b$, to each example $x$. In particular For $c \in C$, define $c'(x, b) = c(x)$ so that the classifiers cannot depend on $b$. This does not change the VC dimension of the classifiers. However, now, any distribution over $\mu$ can be converted to a consistent distribution $\mu'$ over $X'$ whose marginal distribution over $X$ agrees with $\mu$, by making

$$\mu'((x, b), y) = \begin{cases} \mu(x, y) & \text{if } b = y \\ 0 & \text{otherwise.} \end{cases}$$

In other words, $\Pr_{(x,b),y \sim \mu'}[b = y] = 1$. Now, clearly $\mu'$ is consistent. The statement of the lemma applied to $\mu'$ implies the corresponding statement for $\mu$ since the classifiers do not depend on $b$. $\quad \square$

We now prove Lemma B.3.

*Proof of Lemma B.3.* Fix any $n \in \mathbb{N}$, any distribution $P$ over a domain $X$ and any $\beta \in [n]$. Imagine selecting $\mathbf{x}, \mathbf{z} \sim P^n$ by first randomly choosing $2n$ random elements $\mathbf{w} \sim P^{2n}$ and then randomly dividing these elements into two equal sized sequences $\mathbf{x}, \mathbf{z}$. Let $\pi(\mathbf{w})$ denote the distribution over the $(2n)!$ such divisions $\mathbf{x}, \mathbf{z} \in X^n$. For any $g \in G$ and $\mathbf{w} \in X^{2n}$, we claim:

$$\Pr_{(\mathbf{x},\mathbf{z}) \sim \pi(\mathbf{w})} \left[ \sum_i g(x_i) = 0 \ \wedge \ \sum_i g(z_i) \geq \lceil \epsilon n \rceil \right] \leq 2^{-\lceil \epsilon n \rceil}.$$

To see this, suppose $s = \sum_i g(w_i) \geq \epsilon n$ (otherwise the probability above is zero). The probability that all of them are in the test set is at most $2^{-s} \leq 2^{-\epsilon n}$ because the chance that the first rejection is placed in the test set is $1/2$, the second is $(n-1)/(2n-1) < 1/2$, and so forth. The above equation directly implies eq. (17) by dividing by $n$.

We now move to eq. (18). Consider random variables $A = \sum g(x_i)$ and $B = \sum g(z_i)$. It suffices to show that that $B > (1+\alpha)A + r$ with probability $\leq e^{-2\alpha(2+\alpha)^{-2}r}$ for $r = \epsilon n$. Note that since $B = s - A$,

$$B \geq (1+\alpha)A + r \iff A \leq \frac{s-r}{2+\alpha}.$$

Hence, it suffices to prove that

$$\Pr\left[A \leq \frac{s-r}{2+\alpha}\right] \leq e^{-\frac{2\alpha}{(2+\alpha)^2}r}. \tag{34}$$

Let $\mathcal{D}$ be the Bernoulli distribution on $\{0, 1\}$ with mean $\mu = \frac{s}{2n}$. Note that by linearly of expectation, $\mathbb{E}[A] = \mathbb{E}[B] = \mu n$. It is well-known that the probabilities of such an unbalanced split are smaller for sampling without replacement than with replacement [see, e.g., Bardenet et al., 2015]. Thus, it suffices to prove Equation (34) assuming $A$ was sampled by sampling $n$ iid elements $(A_1, \ldots, A_n) \sim \mathcal{D}^n$, and setting $A = \sum_{i=1}^{n} A_i$. By the multiplicative Chernoff bound, for every $\rho \in [0, 1]$,

$$\Pr[A \leq (1-\rho)\mu n] \leq e^{-\rho^2 \mu n/2} = e^{-\rho^2 s/4}.$$

In particular, for $\rho = \frac{\alpha s + 2r}{s(2+\alpha)}$, since $1 - \rho = \frac{2s - 2r}{s(2+\alpha)}$ and $\mu n = s/2$, this gives:

$$\Pr\left[A \leq \frac{s-r}{2+\alpha}\right] \leq e^{-\frac{(\alpha s + 2r)^2}{4(2+\alpha)^2 s}}$$

Hence, it remains to show that the RHS above is at most $\exp\left(-\frac{2\alpha}{(2+\alpha)^2}r\right)$, or equivalently,

$$\frac{(\alpha s + 2r)^2}{4(2+\alpha)^2 s} \geq \frac{2\alpha r}{(2+\alpha)^2}.$$

After multiplying both sides by $4(2+\alpha)^2 s$, the above can be rewritten as $(\alpha s + 2r)^2 \geq 8\alpha sr$, and equivalently as $(\alpha s - 2r)^2 \geq 0$, which indeed always holds. $\qquad\square$

We are now ready to prove Lemma A.1.

*Proof of Lemma A.1.* Note that for $\mathsf{dis}_{h,\mathbf{c}}$ defined as in eq. (29), $\mathsf{dis}_{h,\mathbf{c}}(x) = 0$ if and only if $x \in S(h, \mathbf{c})$. Thus, $\mathbb{1}_{\mathbf{x}}(S(h, \mathbf{c})) = 0$ implies that

$$\sum_{i=1}^{n} \mathsf{dis}_{h,\mathbf{c}}(x_i) = 0. \tag{35}$$

Also note that,

$$\mathbb{1}_{\mathbf{z}}(S(h, \mathbf{c})) = \frac{1}{n}\sum_{i=1}^{n} \mathsf{dis}_{h,\mathbf{c}}(z_i).$$

Hence, it suffices to show

$$\Pr_{\mathbf{x},\mathbf{z} \sim P^n}\left[\exists \mathbf{c} \in C^T, h \in C : \left(\frac{1}{n}\sum_{i=1}^{n}\mathsf{dis}_{h,\mathbf{c}}(x_i) = 0\right) \wedge \left(\frac{1}{n}\sum_{i=1}^{n}\mathsf{dis}_{h,\mathbf{c}}(z_i) > \epsilon\right)\right] \leq \delta \tag{36}$$

By Lemma B.3,

$$\Pr_{\mathbf{x},\mathbf{z} \sim P^n}\left[\exists \phi \in \mathsf{Dis}_T : \left(\sum_{i=1}^{n}\phi(x_i) = 0\right) \wedge \left(\sum_{i=1}^{n}\phi(z_i) \geq \epsilon n\right)\right] \leq 2^{-\epsilon n}\Pi_{\mathsf{Dis}_T}[2n]. \tag{37}$$

Lemma E.1 states that $\Pi_{\mathsf{Dis}_T}[2n] \leq (2n)^{d(T+1)}$ which combined with our choice of $\epsilon$, gives:

$$2^{-\epsilon n}\Pi_{\mathsf{Dis}_T}[2n] \leq 2^{-\epsilon n}(2n)^{d(T+1)} = \delta.$$

Hence, eq. (37) implies eq. (36) because $\mathsf{dis}_{h,\mathbf{c}} \in \Pi_{\mathsf{Dis}_T}$. $\qquad\square$

We now prove Lemma D.6.

*Proof of Lemma D.6.* Note that for $\mathsf{dis}_{h,\mathbf{c}}$ defined as in eq. (29), $\mathsf{dis}_{h,\mathbf{c}}(x) = 1$ if and only if Rejectron rejects $x$ when the algorithm's choices are $h \in C$ and $\mathbf{c} \in C^T$, i.e., $x \notin S$. Thus,

$$\mathbb{1}_{\mathbf{x}}(S) = \frac{1}{n}\sum_{i=1}^{n} \mathsf{dis}_{h,\mathbf{c}}(x_i) \text{ and } \mathbb{1}_{\mathbf{z}}(S) = \frac{1}{n}\sum_{i=1}^{n} \mathsf{dis}_{h,\mathbf{c}}(z_i).$$

Hence, it suffices to show,

$$\Pr_{\mathbf{x},\mathbf{z}\sim P^n}\left[\exists \mathbf{c} \in C^T, h \in C : \frac{1}{n}\sum_{i=1}^{n}\mathsf{dis}_{h,\mathbf{c}}(z_i) > \frac{2}{n}\sum_{i=1}^{n}\mathsf{dis}_{h,\mathbf{c}}(z_i) + \epsilon\right] \le \delta \tag{38}$$

Lemma B.3 (with $\alpha = 1$) implies that:

$$\Pr_{\mathbf{x},\mathbf{z}\sim P^n}\left[\exists \phi \in \mathsf{Dis}_T : \left(\sum_i \phi(x_i) = 0\right) \wedge \left(\sum_i \phi(z_i) \ge \epsilon n\right)\right] \le e^{-\frac{2}{9}\epsilon n}\Pi_{\mathsf{Dis}_T}[2n].$$

Lemma E.1 states that $\Pi_{\mathsf{Dis}_T}[2n] \le (2n)^{d(T+1)}$ which combined with our choice of $\epsilon$, gives:

$$e^{-\frac{2}{9}\epsilon n}\Pi_{\mathsf{Dis}_T}[2n] \le e^{-\frac{2}{9}\epsilon n}(2n)^{d(T+1)} = \delta.$$

Hence, the above implies eq. (38) because $\mathsf{dis}_{h,\mathbf{c}} \in \Pi_{\mathsf{Dis}_T}$. $\qquad\square$

We now prove Lemma A.3.

*Proof of Lemma A.3.* Fix $f, h \in C$. For every $\mathbf{c} \in C^T$, let $S = S(h, \mathbf{c})$ and define:

$$g_{\mathbf{c}}(x) := \begin{cases} 1 & \text{if } f(x) \ne h(x) \wedge x \in S \\ 0 & \text{otherwise.} \end{cases} \text{ and } G := \{g_{\mathbf{c}} : \mathbf{c} \in C^T\}$$

So $G$ depends on $h, f$ which we have fixed. Note that $g_{\mathbf{c}}(x) = 1$ iff $h|_S(x) = 1 - f(x)$. Hence,

$$\frac{1}{n}\sum_{i\in[n]} g_{\mathbf{c}}(\tilde{x}_i) = \mathsf{err}_{\tilde{\mathbf{x}}}(h|_S, f).$$

Equation (32) of Lemma E.2 (with $\gamma = 1/2$ and substituting $Q$ for $P$ and $\epsilon' = 2\epsilon$ for $\epsilon$) implies that for the entire class of functions $G$:

$$\Pr_{\tilde{\mathbf{x}}\sim Q^n}\left[\exists g \in G : \left(\mathbb{E}_{x'\sim Q}[g(x')] > 2\epsilon\right) \wedge \left(\frac{1}{n}\sum_{i\in[n]} g(\tilde{x}_i) \le \epsilon\right)\right] \le 8\Pi_G[2n]e^{-\epsilon n/8}.$$

By definition of $G$, the above implies that,

$$\Pr_{\tilde{\mathbf{x}}\sim Q^n}\left[\exists \mathbf{c} \in C^T : \left(\mathsf{err}_Q(h|_{S(h,\mathbf{c})}, f) > 2\epsilon\right) \wedge \left(\mathsf{err}_{\tilde{\mathbf{x}}}(h|_{S(h,\mathbf{c})}, f) \le \epsilon\right)\right] \le 8\Pi_G[2n]e^{-\epsilon n/8}.$$

Thus, it remains to prove that
$$8\Pi_G[2n]e^{-\epsilon n/8} \le \delta.$$
To bound $\Pi_G[2n]$, note that $g_{\mathbf{c}}(x) = 1 - \mathsf{dis}_{\tilde{f},h,\mathbf{c}}(x)$ and thus $\Pi_G[2n] = \Pi_{\mathsf{Dis}_{\tilde{f},h,T}}[2n]$, which is at most $(2n)^{dT}$ by Lemma E.1. Since $T \le 1/\epsilon$:

$$8(2n)^{dT}e^{-\epsilon n/8} \le 8(2n)^{d/\epsilon}e^{-\epsilon n/8}.$$

Hence it suffices to show that the above is at most $\delta$ for $\epsilon \ge \frac{8\ln 8/\delta}{n} + \sqrt{\frac{8d\ln 2n}{n}}$, or equivalently that,

$$\epsilon\frac{n}{8} - \frac{d}{\epsilon}\ln 2n \ge \ln\frac{8}{\delta}.$$

By multiplying both sides of the equation by $\epsilon \cdot \frac{8}{n}$ we get

$$\epsilon^2 - \frac{8}{n}d\ln 2n \ge \epsilon\frac{8}{n}\ln\frac{8}{\delta}.$$

Substituting $c = \frac{8d \ln 2n}{n}$ and $b = \frac{8 \ln 8/\delta}{n}$, we have $\epsilon \geq b + \sqrt{c}$, and what we need to show above is equivalent to:

$$\epsilon^2 - c \geq \epsilon b$$

or equivalently

$$\epsilon(\epsilon - b) \geq c$$

which holds for any $\epsilon \geq b + \sqrt{c}$ because the LHS above is $\geq (b + \sqrt{c})\sqrt{c} \geq c$. $\qquad\square$

We next prove Lemma A.4.

*Proof of Lemma A.4.* Fix any $T \geq 1$ and any $h \in C$. Consider $\mathrm{dis}_{h,\mathbf{c}}$ as defined in eq. (27), where $\mathrm{dis}_{h,\mathbf{c}}(x) = 1$ iff $x \notin S(h, \mathbf{c})$ is rejected. Thus,

$$\blacksquare_{\mathbf{x}}(S(h, \mathbf{c})) = \frac{1}{n} \sum_{i=1}^{n} \mathrm{dis}_{h,\mathbf{c}}(x_i) \text{ and } \blacksquare_P(S(h, \mathbf{c})) = \mathop{\mathbb{E}}_{x' \sim P}[\mathrm{dis}_{h,\mathbf{c}}(x')].$$

By Lemma E.2 (Equation (31)), the probability that any such $\mathrm{dis}_{h,\mathbf{c}} \in \mathrm{Dis}_T$ is 0 on all of $\mathbf{x}$ but has expectation on $P$ greater than $\xi = \frac{2}{n}(d(T + 1) \ln(2n) + \ln \frac{2}{\delta})$ is at most:

$$2\Pi_{\mathrm{Dis}_T}[2n]2^{-\xi n/2} \leq 2(2n)^{d(T+1)}2^{-\xi n/2} = \delta,$$

where the first inequality follows from the fact that $\Pi_{\mathrm{Dis}_T}[2n] \leq (2n)^{d(T+1)}$, which follows from Lemma E.1. Similarly, eq. (32) of Lemma E.2 (with $\gamma = 1/2$ and $\epsilon = 2\alpha$) implies that:

$$\Pr_{\mathbf{x} \sim P^n}\left[\exists h, \mathbf{c} : \left(\mathop{\mathbb{E}}_{x' \sim P}[\mathrm{dis}_{h,\mathbf{c}}(x')] > 2\alpha\right) \wedge \left(\frac{1}{n}\sum_{i \in [n]} \mathrm{dis}_{h,\mathbf{c}}(x_i) \leq \alpha\right)\right] \leq 8\Pi_{\mathrm{Dis}_T}[2n]e^{-\alpha n/8}.$$

For $\alpha$ as in the lemma, the right hand side above is at most $\delta$. $\qquad\square$

# F  Proofs of lower bounds

We note that, in the lower bound of Theorem 4.4, the distribution $Q$ is fixed, independent of $f$. Since $Q$ is used only for unlabeled test samples, the learning algorithm can gain no information about $Q$ even if it is given a large number $m$ of test samples. In particular, it implies that even if one has $n$ training samples and infinitely many samples from $Q$, one cannot achieve error less than $\Omega(\sqrt{d/n})$. It would be interesting to try to improve the lower-bound to have a specific dependence on $m$ (getting $\Omega(\sqrt{1/n} + 1/m)$ is likely possible using a construction similar to the one below). Also, the lower-bound could be improved if one had fixed distributions $\nu, P, Q$ independent of $n$.

*Proof of Theorem 4.4.* Let $X = \mathbb{N}$ and $C$ be the concept class of functions which are 1 on exactly $d$ integers, which can easily be seen to have VC dimension $d$. The distribution $P$ is simply uniform over $[8n] = \{1, 2, \ldots, 8n\}$. Let $k = \sqrt{8dn}$. The distribution $Q$ is uniform over $[k]$. We consider a distribution $\nu$ over functions $f \in C$ that is uniform over the $\binom{k}{d}$ functions that are 1 on exactly $d$ points in $[k]$. We will show,

$$\mathbb{E}_{f \sim \nu}\left[\mathbb{E}_{\substack{\mathbf{x} \sim P^n \\ \tilde{\mathbf{x}} \sim Q^n}}\left[\blacksquare_P + \mathrm{err}_Q\right]\right] \geq K\sqrt{\frac{d}{n}}. \tag{39}$$

By the probabilistic method, this would imply the lemma.

The set of training samples is $T = \{x_i : i \in [n]\} \subseteq [8n]$. Say an $j \in [k]$ is "unseen" if it does not occur as a training example, $j \notin T$. WLOG, we may assume that the learner makes the same classification $h|_S$ for each unseen $j \in [k]$ since an asymmetric learner can only be improved by making the (same) optimal decision for each unseen $j \in [k]$, where the optimal decisions are defined to be those that minimize $\mathbb{E}[\blacksquare_P + \mathrm{err}_Q \mid \mathbf{x}, f(\mathbf{x})]$. (The unlabeled test are irrelevant because $Q$ is fixed.)

Now, let $U \leq k$ be the random variable that is the number of seen $j \in [k]$ and $V \leq d$ be the number that are labeled 1 (which the learner can easily determine).

$$U = |T \cap [k]|$$
$$V = |\{j \in T \cap [k] : f(j) = 1\}|.$$

Note that $\mathbb{E}[U] \leq k/8$ and $\mathbb{E}[V] \leq d/8$ since each $j \in [k]$ is observed with probability $\leq 1/8$ by choice of $P$ (the precise observation probability is $1 - (1 - \frac{1}{8n})^n \leq \frac{1}{8}$). These two inequalities implies that,

$$\mathbb{E}\left[\frac{U}{k} + \frac{V}{d}\right] \leq \frac{1}{8} + \frac{1}{8} = \frac{1}{4}.$$

Thus, by Markov's inequality,

$$\Pr\left[\frac{U}{k} + \frac{V}{d} \leq \frac{1}{2}\right] \geq \frac{1}{2}.$$

This implies that, with probability $\geq 1/2$, both $U \leq k/2$ *and* $V \leq d/2$. Suppose this event happens. Now, consider three cases.

Case 1) if the learner predicts $\blacksquare$ on all unseen $j \in [k]$, then

$$\blacksquare_P \geq \frac{k}{2} \cdot \frac{1}{8n} = \sqrt{\frac{d}{32n}}$$

because there are at least $k/2$ unseen $j \in [k]$ and each has probability $\frac{1}{8n}$ under $P$.

Case 2) if the learner predicts 0 on all unseen $j \in [k]$, then

$$\mathrm{err}_Q \geq \frac{d}{2} \cdot \frac{1}{k} = \sqrt{\frac{d}{32n}},$$

because there are at least $d/2$ 1's that are unseen and each has probability $1/k$ under $Q$.

Case 3) if the learner predicts 1 on all unseen $j \in [k]$ then

$$\mathrm{err}_Q \geq \left(\frac{k}{2} - d\right)\frac{1}{k} = \frac{1}{2} - \sqrt{\frac{d}{8n}} \geq \sqrt{\frac{d}{8n}} > \sqrt{\frac{d}{32n}}$$

because there are at least $k/2 - d$ unseen 0's, each with probability $1/k$ under $Q$ (and by assumption $n \geq 2d$ so $\sqrt{d/(8n)} \leq 1/4$). Thus in all three cases, $\blacksquare_P + \mathrm{err}_Q \geq \sqrt{d/(32n)}$. Hence,

$$\mathbb{E}\left[\blacksquare_P + \mathrm{err}_Q \mid U \leq k/2, V \leq d/2\right] \geq \sqrt{\frac{d}{32n}}$$

Since $U \leq k/2, V \leq d/2$ happens with probability $\geq 1/2$, we have that $\mathbb{E}[\blacksquare_P + \mathrm{err}_Q] \geq \frac{1}{2}\sqrt{d/(32n)}$ as required. This establishes eq. (39). $\qquad\square$

We now prove our agnostic lower bound.

*Proof of Lemma D.1.* Let $X = \mathbb{N}$ and $C$ consist of the singleton functions that are 1 at one integer and 0 elsewhere. The VC dimension of $C$ is easily seen to be 1.

Consider first the case in which $\tilde{\eta} \geq \sqrt{\eta/8}$. In this case, we must construct distributions $\mu, \tilde{\mu}$ and $f \in C$ such that, $\mathbb{E}[\mathrm{err}_{\tilde{\mu}}(h|_S) + \blacksquare_\mu(S)] \geq \tilde{\eta}$. This is trivial: let $\mu$ be arbitrary and $\tilde{\mu}(1,1) = \tilde{\eta}$ and $\tilde{\mu}(1,0) = 1 - \tilde{\eta}$. It is easy to see that no classifier has error less than $\tilde{\eta}$ since $\tilde{\eta} \leq 1/2$.

Thus it suffices to give $\mu, \tilde{\mu}$ and $f \in C$ such that, $\mathrm{err}_{\tilde{\mu}}(f) = 0$, $\mathrm{err}_\mu(f) = \eta$, and,

$$\mathop{\mathbb{E}}_{\substack{(\mathbf{x},\mathbf{y}) \sim \mu^n \\ (\tilde{\mathbf{x}},\tilde{\mathbf{y}}) \sim \tilde{\mu}^n}}[\mathrm{err}_{\tilde{\mu}}(h|_S) + \blacksquare_\mu(S)] \geq \sqrt{\eta/8}. \tag{40}$$

In particular, we will give a distribution over $f, \mu, \tilde{\mu}$ for which the above holds for the output $h|_S$ of any learning algorithm. By the probabilistic method, this implies that for each learning algorithm, there is at least $f, \mu, \tilde{\mu}$ for which eq. (40) holds. To this end, let $k = \lfloor \sqrt{2/\eta} \rfloor$. Let $\mu$ be the distribution

which has $\mu(x,0) = \eta/2$ for $x \in [k]$ and $\mu(k+1,0) = 1 - k\eta/2$, so $\mu$ has $y = 0$ with probability 1. Let $f$ be 1 for a uniformly random $x^* \in [k]$ so $\text{err}_\mu(f) = \eta/2$. Let $\tilde\mu$ be the distribution where $\tilde\mu(x, f(x)) = 1/k$ for $x \in [k]$, so $x$ is uniform over $[k]$ with $\text{err}_{\tilde\mu}(f) = 0$.

Now, given the above distribution over $f, \mu, \tilde\mu$, there is an optimal learning algorithm that minimizes $\mathbb{E}[\text{err}_{\tilde\mu} + \blacksquare_\mu]$. Moreover, notice that the algorithm learns nothing about $\mu$ or $\tilde\mu$ from the training data since $\mu$ is fixed as is the distribution over unlabeled examples. Thus the optimal learner, by symmetry, may be taken to make the same classification for all $x \in [k]$. Thus, consider three cases.

- The algorithm predicts $h|_S(x) = \blacksquare$ for all $x \in [k]$. In this case,

$$\blacksquare_\mu \geq k\frac{\eta}{2} = \lfloor \sqrt{2/\eta} \rfloor \frac{\eta}{2} \geq \frac{1}{2}\sqrt{\eta/2}$$

  using the fact that $\lfloor r \rfloor \geq r/2$ for $r \geq 1$.

- The algorithm predicts $h|_S(x) = 0$ for all $x \in [k]$. In this case,

$$\text{err}_{\tilde\mu} = \frac{1}{k} \geq \sqrt{\eta/2}.$$

- The algorithm predicts $h|_S(x) = 1$ for all $x \in [k]$. In this case, since $\eta \leq 1/2$, $k \geq 2$ and $\text{err}_{\tilde\mu} \geq 1/2$.

In all three cases, $\text{err}_{\tilde\mu} + \blacksquare_\mu \geq \sqrt{\eta/8}$ proving the lemma. $\qquad\square$

We now present the theorem and proof of our transductive lower bound.

**Theorem F.1** (Transductive lower bound). *There exists a constant $K > 0$ such that: for any $d \geq 1$ there exists a concept class of VC dimension $d$ where, for any $m, n \geq 4d$ there exists a distribution $P$, and an adversary $\mathcal{A} : X^{n+m} \to X^m$, such that for any learner $L : X^n \times Y^n \times X^m \to Y^X \times 2^X$ there is a function $f \in C$ such that:*

$$\mathbb{E}_{\substack{\mathbf{x} \sim P^n \\ \mathbf{z} \sim P^m}}[\blacksquare_{\mathbf{z}} + \text{err}_{\tilde{\mathbf{x}}}] \geq K\sqrt{\frac{d}{\min\{m,n\}}}$$

*where $\tilde{\mathbf{x}} = \mathcal{A}(\mathbf{x}, \mathbf{z})$ and $h|_S = L(\mathbf{x}, f(\mathbf{x}), \tilde{\mathbf{x}})$.*

*Proof of Theorem F.1.* Just as in the proof of Theorem 4.4, let $X = \mathbb{N}$ and $C$ again be the concept class of functions that have exactly $d$ 1's, which has VC dimension $d$. Again, let $P$ be the uniform distribution over $[N]$ for $N = 8n$.

We will construct a distribution $\nu$ over $C$ and randomized adversary $\mathcal{A}(\mathbf{x}, \mathbf{z}, f)$ that outputs $\tilde{\mathbf{x}} \in X^n$ such that, for all $L$,

$$\mathbb{E}[\blacksquare_{\mathbf{z}} + \text{err}_{\tilde{\mathbf{x}}}] \geq \lambda,$$

where $\lambda$ is a lower bound and expectations are over $\mathbf{x} \sim P^n, \mathbf{z} \sim P^m$ and $f \sim \nu$. By the probabilistic method again, such a guarantee implies that for any learner $L$, there exists some $f \in C$ and deterministic adversary $\mathcal{A}(\mathbf{x}, \mathbf{z})$ where the above bound holds for that learner.

We will show two lower bounds that together imply the lemma. The first lower bound will follow from Theorem 4.4 and show that,

$$\mathbb{E}[\blacksquare_{\mathbf{z}} + \text{err}_{\tilde{\mathbf{x}}}] \geq K\sqrt{d/n},$$

where expectations are over $\mathbf{x} \sim P^n, \mathbf{z} \sim P^m, f \sim \nu$. Here $K$ is the constant from Theorem 4.4. To get this, the adversary $\mathcal{A}(\mathbf{x}, \mathbf{z}, f)$ simply ignores the true tests $\mathbf{z}$ and selects $\tilde{\mathbf{x}} \sim Q^m$. By linearity of expectation, for any learner, $\mathbb{E}[\blacksquare_{\mathbf{z}}] = \mathbb{E}[\blacksquare_P]$ and $\mathbb{E}[\text{err}_{\tilde{\mathbf{x}}}] = \mathbb{E}[\text{err}_Q]$.

It remains to show a distribution $\nu$ over $C$ and adversary $A$ such that, for all learners,

$$\mathbb{E}[\blacksquare_{\mathbf{z}} + \text{err}_{\tilde{\mathbf{x}}}] \geq K\sqrt{d/m}, \tag{41}$$

for some constant $K$ and $m < n$ (for $m \geq n$, the previous lower bound subsumes this). Let $\nu$ be the uniform distribution over those $f \in C$ that have all $d$ 1's in $[N]$, i.e., uniform over $\{f \in C : \sum_{i \in [N]} f(i) = d\}$.

Let $A := \{x \in [N] : f(x) = 0\}$ and $B := \{x \in \mathbb{N} : f(x) = 1\}$ so $|A| = N - d$ and $|B| = d$.

Let $a = \lfloor \sqrt{md} \rfloor$ and $b = \lceil d/2 \rceil$, and $r = \lfloor m/(a+b) \rfloor$. The adversary will try to construct a dataset $\tilde{\mathbf{x}}$ with the following properties:

- $\tilde{\mathbf{x}}$ contains exactly $a$ distinct $\tilde{x} \in A$ and each has exactly $r$ copies. (Since $a \leq m < N - d$, this is possible.)

- There are exactly $b$ distinct $\tilde{x} \in B$ and each has exactly $r$ copies.

- The remaining $m - r(a+b)$ examples are all at $\tilde{x} = N + 1$ (these are "easy" as the learner can just label them 0 if it chooses).

We say $x$ is *seen* if $x \in \mathbf{x}$ (this notation indicates $x \in \{x_i : i \in [n]\}$ did not occur in the training set) and *unseen* otherwise. Now, we first observe that with probability $\geq 1/8$, the following event $E$ happens: there are at most $d - b$ seen 1's ($x_i \in B$) in the training set and there are at least $a$ distinct unseen 0's in the true test set $\mathbf{z}$, i.e.,

$$V_1 := |\{i \in [n] : x_i \in B\}| \leq d - b$$
$$V_0 := |\{z \in A : (z \in \mathbf{z}) \wedge (z \notin \mathbf{x})\}| \geq a$$

Note that $\mathbb{E}[V_1] = dn/N = d/8$. Markov's inequality guarantees that with probability $\geq 3/4$, $V_1 \leq d/2$ (otherwise $\mathbb{E}[V_1] > d/8$). Since $V_1$ is integer, this means that with probability $\geq 3/4$, $V_1 \leq \lfloor d/2 \rfloor = d - b$. Similarly, for any $i \in A$, the probability that it occurs in $\mathbf{z}$ and not in $\mathbf{x}$ is,

$$\left(1 - \frac{1}{N}\right)^n \left(1 - \left(1 - \frac{1}{N}\right)^m\right) \geq \left(1 - \frac{n}{N}\right)\left(1 - e^{-\frac{m}{N}}\right) \geq \frac{7}{8} \cdot \frac{15}{16}\frac{m}{N} \geq 0.8\frac{m}{N},$$

where in the above we have used the fact that $(1 - t) \leq e^{-t}$ for $t > 0$ and $1 - e^{-t} \geq (15/16)t$ for $t \leq 1/8$. Hence, since $|A| = N - d$,

$$\mathbb{E}[V_0] \geq (N - d)0.8\frac{m}{N} \geq \left(\frac{7}{8}N\right)0.8\frac{m}{N} = 0.7m \geq 0.7a.$$

In particular, Markov's inequality implies that with probability at least 0.4, $V_0 \geq 0.5m$ (otherwise $\mathbb{E}[V_0] < 0.6(0.5m) + 0.4m = 0.7m$). Thus, with probability $\geq 1 - 1/4 - 0.6 \geq 1/8$.

If this event $E$ does not happen, then the adversary will take all $\tilde{x} = N + 1$, making learning easy. However, if $E$ does happen, then there must be at least $a$ unseen 0's in $\mathbf{z}$ and $b$ unseen 1's and the the adversary will select $a$ random unseen 0's from $\mathbf{z}$ and $b$ random unseen 1's, uniformly at random. It will repeat these examples $r$ times each, add $m - r(a+b)$ copies of $\tilde{x} = N + 1$, and permute the $m$ examples.

Now that the adversary and $v$ have been specified, we can consider a learner $L$ that minimizes the objective $\mathbb{E}[\blacksquare_{\mathbf{z}} + \mathrm{err}_{\tilde{\mathbf{x}}}]$. Clearly this learner may reject $N + 1 \notin S$ as this cannot increase the objective. Now, by symmetry the learner may also be assumed to make the same classification on all $r(a+b)$ examples $\tilde{x} \in [N]$ as these examples are all unseen and indistinguishable since $B$ is uniformly random.

Case 1) If $h|_S(\tilde{\mathbf{x}}_i) = \blacksquare$ for all $i$ then

$$\blacksquare_{\mathbf{z}} = \frac{a}{m} = \frac{\lfloor \sqrt{md} \rfloor}{m} \geq \frac{\sqrt{md}/2}{m} = \frac{1}{2}\sqrt{\frac{d}{m}},$$

using the fact that $a \geq \sqrt{md}/2$ because $a \geq \sqrt{md}/2$ since $\lfloor t \rfloor \geq t/2$ for $t \geq 1$.

Case 2) If $h|_S(\tilde{\mathbf{x}}_i) = 0$ for all $i$ then,

$$\mathrm{err}_{\tilde{\mathbf{x}}} = \frac{br}{m} \geq \frac{b\sqrt{m/d}}{4m} = \frac{b}{4\sqrt{md}} \geq \frac{d}{8\sqrt{md}} = \frac{1}{8}\sqrt{\frac{d}{m}}$$

In the above we have used the fact $b \geq d/2$ and that $r \geq \frac{1}{4}\sqrt{m/d}$, which can be verified by noting that:

$$\frac{m}{a+b} \geq \frac{m}{2a} \geq \frac{m}{2\sqrt{md}} = \frac{1}{2}\sqrt{\frac{m}{d}} \geq 1$$

and hence $r \geq \lfloor m/(a+b) \rfloor \geq \frac{1}{2} m/(a+b) \geq \frac{1}{4}\sqrt{m/d}$ again since $\lfloor t \rfloor \geq t/2$ for $t \geq 1$.

Case 3) If $h|_S(\tilde{\mathbf{x}}_i) = 0$ for all $i$ then, since $b \leq a$

$$\mathrm{err}_{\tilde{\mathbf{x}}} = \frac{b}{a+b} \geq \frac{1}{2}.$$

In all three cases, we have,

$$\blacksquare_{\mathbf{z}} + \mathrm{err}_{\tilde{\mathbf{x}}} \geq \frac{1}{8}\sqrt{\frac{d}{m}}.$$

Since $E$ happens with probability $\geq 1/2$, we have,

$$\mathbb{E}[\blacksquare_{\mathbf{z}} + \mathrm{err}_{\tilde{\mathbf{x}}}] \geq \Pr[E]\,\mathbb{E}[\blacksquare_{\mathbf{z}} + \mathrm{err}_{\tilde{\mathbf{x}}} \mid E] \geq \frac{1}{8} \cdot \frac{1}{8}\sqrt{\frac{d}{m}}.$$

This is what was required for eq. (41). $\qquad\square$

## G  Tight bounds relating train and test rejections

**Lemma G.1.** *For any $S \subseteq X$ and distributions $P, Q$ over $X$:*

$$\blacksquare_Q(S) \leq \blacksquare_P(S) + |P - Q|_{\mathrm{TV}}. \tag{42}$$

*Further, for any $\Lambda \geq 0$,*

$$\Pr_{x \sim Q}\left[x \notin S \ \text{and}\ Q(x) \leq \Lambda P(x)\right] \leq \Lambda\,\blacksquare_P(S). \tag{43}$$

*Proof of Lemma G.1.* For eq. (42), note that one can sample a point from $\tilde{x} \sim Q$ by first sampling $x \sim P$ and then changing it with probability $|P - Q|_{\mathrm{TV}}$. This follows from the definition of total variation distance. Thus, the probability that $\tilde{x}$ is rejected is at most the probability $x$ is rejected plus the probability $x \neq \tilde{x}$, establishing eq. (42). To see eq. (43), note

$$\Pr_{x \sim Q}\left[x \notin S \ \text{and}\ Q(x) \leq \Lambda P(x)\right] = \sum_{x \in \bar{S}:Q(x) \leq \Lambda P(x)} Q(x) \leq \sum_{x \in \bar{S}:Q(x) \leq \Lambda P(x)} \Lambda P(x).$$

Clearly the above is at most $\sum_{x \in \bar{S}} \Lambda P(x) = \Lambda\,\blacksquare_P$. $\qquad\square$

We now move on to tightly relating test and training rejections. As motivation, note that if one knew $P$ and $Q$, it would be natural to take $S^* := \{x \in X : Q(x) \leq P(x)/\epsilon\}$ for some $\epsilon > 0$. For $x \notin S^*$, i.e., $x \in \bar{S}^*$, $P(x) < \epsilon Q(x)$. This implies that $\blacksquare_P(S^*) = P(\bar{S}^*) < \epsilon$. It is also straightforward to verify that $\mathrm{err}_Q(h|_{S^*}) \leq \mathrm{err}_P(h)/\epsilon$. This means that if one can find $h$ of error $\epsilon^2$ on $P$, e.g., using a PAC-learner, then this gives,

$$\blacksquare_P(S^*) + \mathrm{err}_Q(h|_S^*) \leq 2\epsilon.$$

This suggests that perhaps we could try to learn $P$ and $Q$ and approximate $S^*$. Unfortunately, this is generally impossible—one cannot even distinguish the case where $P = Q$ from the case where $P$ and $Q$ have disjoint supports with fewer than $\Omega(\sqrt{|X|})$ examples.[6]

While we cannot learn $S^*$ in general, these sets $S^*$ do give the tightest bounds on $\blacksquare_Q$ in terms of $\blacksquare_P$.

**Lemma G.2.** *For any $S \subseteq X$ and distributions $P, Q$ over $X$ and any $\epsilon \geq 0$ such that $\blacksquare_P(S) \leq \blacksquare_P(S^*)$,*

$$\blacksquare_Q(S) \leq \blacksquare_Q(S^*). \tag{44}$$

Note that the $\blacksquare_Q(S) \leq \blacksquare_P(S) + |P - Q|_{\mathrm{TV}}$ bound can be much looser than the bound in the above lemma. For example, $|P - Q|_{\mathrm{TV}} = 0.91$ yet $\blacksquare_Q(S^*) = 0.1$ for $X = \{0, 1, \ldots, 100\}$, $P$ uniform over $\{1, \ldots, 100\}$, $Q$ uniform over $\{0, 1, \ldots, 9\}$, $\blacksquare_P(S) = 0$, and $\epsilon = 0.1$ (since $S^* = \{1, 2, \ldots, 100\}$ and only $0 \notin S^*$). One can think of classifying images of a mushroom as "edible" or not based on training data of 100 species of mushrooms, with test data including one new species.

*Proof.* Since $\epsilon Q(x) - P(x) > 0$ iff $x \notin S^*$,

$$\epsilon\, \mathrm{rej}_Q(S^*) - \mathrm{rej}_P(S^*) = \sum_{x \notin S^*} \epsilon Q(x) - P(x)$$

$$\geq \sum_{x \notin S} \epsilon Q(x) - P(x) = \epsilon\, \mathrm{rej}_Q(S) - \mathrm{rej}_P(S)$$

$$\Rightarrow \quad \epsilon(\mathrm{rej}_Q(S^*) - \mathrm{rej}_Q(S)) \geq \mathrm{rej}_P(S^*) - \mathrm{rej}_P(S) \geq 0.$$

$\square$

# H  Proofs of Auxiliary Lemmas

**Claim H.1.** *Let $f \in C, \epsilon, \delta > 0$, $n, k \geq 1$ and $P, Q_1, \ldots, Q_k$ be distributions over $X$. Given a $\left(\frac{\epsilon}{k+1}, \delta, n\right)$-PQ-learner $L$, $\mathbf{x} \sim P^n$, $f(\mathbf{x})$, and additional unlabeled samples $\mathbf{z} \sim P^n, \tilde{\mathbf{x}}_1 \sim Q_1^n, \ldots, \tilde{\mathbf{x}}_k \sim Q_k^n$, one can generate $\tilde{\mathbf{x}} \in X^n$ such that $h|_S = L(\mathbf{x}, f(\mathbf{x}), \tilde{\mathbf{x}})$ satisfies,*

$$\Pr\left[\mathrm{rej}_P + \mathrm{err}_P + \sum_i \mathrm{err}_{Q_i} \leq \epsilon\right] \geq 1 - \delta.$$

*Proof of Claim H.1.* Let $Q = \frac{1}{k+1}\left(P + Q_1 + \cdots + Q_k\right)$ be the blended distribution. Give $n$ samples from $P$ and each $Q_i$, one can straightforwardly construct $n$ iid samples $\tilde{\mathbf{x}} \sim Q$. Running $L(\mathbf{x}, f(\mathbf{x}), \tilde{\mathbf{x}})$ gives the guarantee that with prob. $\geq 1 - \delta$, $(k+1)(\mathrm{rej}_P + \mathrm{err}_Q) \leq \epsilon$ which implies the claim since $(k+1)\mathrm{err}_Q = \mathrm{err}_P + \sum \mathrm{err}_{Q_i}$. $\square$

*Proof of Lemma 4.1.* To maximize $s_t$ using the ERM oracle for $C$, construct a dataset consisting of each training example, labeled by $h$, repeated $\Lambda$ times, and each test example in $\tilde{x}_i \in S_t$, labeled $1 - h(\tilde{x}_i)$, included just once. Running ERM on this artificial dataset returns a classifier of minimal error on it. But the number of errors of classifier $c$ on this artificial dataset is:

$$\Lambda \sum_{i \in [n]} |c(x_i) - h(x_i)| + \sum_{i : \tilde{x}_i \in S_t} |c(\tilde{x}_i) - (1 - h(\tilde{x}_i))| =$$

$$\Lambda \sum_{i \in [n]} |c(x_i) - h(x_i)| + \sum_{i : \tilde{x}_i \in S_t} 1 - |c(\tilde{x}_i) - h(\tilde{x}_i)|,$$

which is equal to $\left|\{i \in [n] : \tilde{x}_i \in S_t\}\right| - n s_t(c)$. Hence $c$ minimizes error on this artificial dataset if and only if it maximizes $s_t$ of the algorithm.

Next, let $T$ be the number of iterations of the algorithm Rejectron, so its output is $h|_{S_{T+1}}$. We must show that $T \leq \lfloor 1/\epsilon \rfloor$. To this end, note that by definition, for every $t \in [T]$ it holds that $S_{t+1} \subseteq S_t$, and moreover,

$$\frac{1}{n}\left|\{i \in [n] : \tilde{x}_i \in S_t\}\right| - \frac{1}{n}\left|\{i \in [n] : \tilde{x}_i \in S_{t+1}\}\right| = \mathrm{err}_{\tilde{\mathbf{x}}}(h|_{S_t}, c_t) \geq s_t(c_t) > \epsilon. \quad (45)$$

Hence, the fraction of additional rejected test examples in each iteration $t \in [T]$ is greater than $\epsilon$, and hence $T < 1/\epsilon$. Since $T$ is an integer, this means that $T \leq \lfloor 1/\epsilon \rfloor$.

For efficiency, of course each $S_t$ is not explicitly stored since even $S_1 = X$ could be infinite. Instead, note that to execute the algorithm, we only need to maintain: (a) the subset of indices $Z_t = \{j \in [n] \mid \tilde{x}_j \in S_t\}$ of test examples which are in the prediction set, and (b) the classifiers $h, c_1, \ldots, c_T$. Also note that updating $Z_t$ from $Z_{t-1}$ requires evaluating $c_t$ at most $n$ times. In this fashion, membership in $S_t$ and $S = S_{T+1}$ can be computed efficiently and output in a succinct manner. $\square$

# I  Experiments

Rather than classifying sensitive attributes such as explicit images, we perform simple experiments on handwritten letter classification from the popular EMNIST dataset [Cohen et al., 2017]. For both

**Figure 4:** Trade-offs between rejection rate on $P$ and error rate on $Q$. The error on $Q$ (in blue) is the fraction of errors *among selected examples* (unlike $\text{err}_Q$ in our analysis).

**Figure 5:** Adversarial choices of *a d e h l n r t*, misclassified by the Random Forest classifier.

experiments, the training data consisted of the eight lowercase letters *a d e h l n r t*, chosen because they each had more than 10,000 instances. From each letter, 3,000 instances of each letter were reserved for use later, leaving 7,000 examples, each constituting 56,000 samples from $P$.

We then considered two test distributions, $Q_{\text{adv}}, Q_{\text{nat}}$ representing adversarial and natural settings. $Q_{\text{adv}}$ consisted of a mix of 50% samples from $P$ (the 3,000 reserved instances per lower-case letter mentioned above) and 50% samples from an adversary that used a classifier $h$ as a black box. To that, we added 3,000 adversarial examples for each letter selected as follows: the reserved 3,000 letters were labeled by $h$ and the adversary selected the first misclassified instance for each letter. Misclassified examples are shown in Figure 5. It made 3,000 imperceptible modifications of each of the above instances by changing the intensity value of a single pixel by at most 4 (out of 256). The result was 6,000 samples per letter constituting 48,000 samples from $Q_{\text{adv}}$.

For $Q_{\text{nat}}$, the test set also consisted of 6,000 samples per letter, with 3,000 reserved samples from $P$ as above. In this case, the remaining half of the letters were simply upper-case[7] versions of the letters *A D E H L N R T*, taken from the EMNIST dataset (case information is also available in that dataset). Again the dataset size is 48,000. We denote this test distribution by $Q_{\text{nat}}$.

In Figure 4, we plot the trade-off between the rejection rate on $P$ and the error rate on $Q$ of the URejectron algorithm. Since this is a multi-class problem, we implement the most basic form of the URejectron algorithm, with $T = 1$ iterations. Instead of fixing parameter $\Lambda$, we simply train a predictor $h^{\text{Dis}}$ to distinguish between examples from $P$ and $Q$, and train a classifier $h$ on $P$. We trained two models, a random forest (with default parameters from scikit-learn [Pedregosa et al., 2011]) and a neural network. Complete details are provided at the end of this section. We threshold the prediction scores of distinguisher $h^{\text{Dis}}$ at various values. For each threshold $\tau$, we compute the fraction of examples from $P$ that are rejected (those with prediction score less than $\tau$), and similarly for $Q$, and the error rate of classifier $h$ on examples from $Q$ that are *not* rejected (those with prediction score at least $\tau$). We see in Figure 4 that for a suitable threshold $\tau$ our URejectron algorithm achieves both low rejection rate on $P$ and low error rate on $Q$. Thus on these problems the simple algorithm suffices.

We compare to the state-of-the-art SC algorithm SelectiveNet [Geifman and El-Yaniv, 2019]. We ran it to train a selective neural network classifier on $P$. SelectiveNet performs exceptionally on $Q_{\text{nat}}$, achieving low error and reject almost exclusively upper-case letters. It of course errs on all adversarial examples from $Q_{\text{adv}}$, as will all existing SC algorithms (no matter how robust) since they all choose $S$ without using unlabeled test examples.

**Models** A Random Forest Classifier $h_{\text{RF}}$ from Scikit-Learn (default parameters including 100 estimators) [Pedregosa et al., 2011] and a simple neural network $h_{\text{NN}}$ consisting of two convolutional

layers followed by two fully connected layers[8] were fit on a stratified 90%/10% train/test split. The network parameters are trained with SGD with momentum (0.9), weight decay ($2 \times 10^{-4}$), batch size (128), for 85 epochs with a learning rate of 0.1, that is decayed it by a factor of 10 on epochs 57 and 72. $h_{\mathrm{RF}}$ had a 3.6% test error rate on $P$, and $h_{\mathrm{NN}}$ had a 1.3% test error rate on $P$.

**SelectiveNet**    SelectiveNet requires a target coverage hyperparameter which in our experiments is fixed to 0.7. We use an open-source PyTorch implementation of SelectiveNet that is available on GitHub [9], with a VGG 16 architecure [Simonyan and Zisserman, 2015]. To accommodate the VGG 16 architecure without changes, we pad all images with zeros (from 28x28 to 32x32), and duplicate the channels (from 1 to 3). SelectiveNet achieves rejection rates of 21.08% ($P$), 45.89% ($Q_{\mathrm{nat}}$), and 24.04% ($Q_{\mathrm{adv}}$), and error rates of 0.02% ($P$), 0.81% ($Q_{\mathrm{nat}}$), and 76.78% ($Q_{\mathrm{adv}}$).

## Footnotes

[5]In Section 4, we assumed that $\eta = \tilde{\eta}$ for simplicity, yet here we consider the more general case where $\eta$ and $\tilde{\eta}$ may differ.

[6]To see this, consider the cases where $P = Q$ are both the uniform distribution over $X$ versus the case where they are each uniform over a random partition of $X$ into two sets of equal size. By the classic *birthday paradox*, with $O(\sqrt{|X|})$ samples both cases will likely lead to random disjoint sets of samples.

[7]In some datasets, letter classes consist of a mix of upper- and lower-case, while in others they are assigned different classes (EMNIST has both types of classes). In our experiments, they belong to the same class.

[8]`https://github.com/pytorch/examples/blob/master/mnist/main.py`

[9]`https://github.com/pranaymodukuru/pytorch-SelectiveNet`