[Reviews · NeurIPS 2020]

Review 1

Summary and Contributions: The paper proposes the new problem statement of PQ-learning a selective classifier (one which can abstain) and two algorithms (Rejectron and Urejectron) that are PQ-learners of selective classifiers. Informally, PQ learning refers to a problem where the learner gets labeled examples from a distribution P and unlabeled examples from distribution Q and needs to learn a selective classifier in polynomial time that has low error on Q and low rejection rate on P. The main contributions are: * formalizing PQ-learning of a selective classifier * proposing two algorithms (Rejectron and Urejectron) within this framework * providing and proving bounds for these algorithms in terms of errors on Q and rejection rates on P

Strengths: Novelty: the main contributions of the paper (PQ-learning, Rejectron, Urejectron) are novel to me and would clearly justify a publication at NeurIPS in terms of novelty. Significance: generalization and controlled behaviour for out-of-distribution data is one of the core challenges in terms of robust machine learning and the presented framework and algorithms make substantial progress. As outlined below, I am however not fully convinced that the specific setting studied in this paper is of relevance (the authors may convince me of the opposite in their response) Soundness of claims: the definitions, theorems, and proofs are well presented and I have not found any flaws in the material.

Weaknesses: Significance: I am not convinced that the considered setting is relevant: * in the "natural" setting (as in the example of a face detector trained primarily based on light-skinned faces P and not on dark-skinned faces Q), providing unlabeled samples from Q requires (a) the developer being aware that P is not representative (b) being able to collect samples from Q and (c) not label these samples from Q. This seems implausible to me. Particularly, it would be unethical to apply selective classification for reducing labeling cost by skipping step c (as discussed by the authors also in the broader impact section). * in the presence of an adversary, if one assumes that the adversary has access to the classifier h (white-box), then it would also make sense to assume the adversary having access to the set S of non-abstained inputs and select Q_adv from within this set. This would result in the typical arms race, where S would need to be adapted to an adversary that itself chooses Q_adv depending on S. Assuming a fixed adversary unaware of S seems of little relevance. Empirical evaluation: * The empirical evaluation is conducted on a very much simplified version of the proposed algorithms, which consists of "training a predictor h^{Dis} to distinguish between examples from P and Q, and train a classifier h on P". I would consider this rather as a baseline (because it is the naive thing to do in a setting where fixed test examples from distribtion Q are given). I would expect the authors to shows in the empirical evaluation how the proposed algorithms perform relative to this baseline, that is Urejectron with T>1 and actually using a threshold \Lambda. * The way Q_adv is constructed is not a challenging setting because it consists of just very many (3,000) imperceptible modifications of a few examples. These can of course very easily be identified and be abstained. Evaluation on more challengingsettings would be desirable, for instance: create a set of misclassified examples and create Q_adv such that it consists of convex combinations of pairs of these misclassified examples.

Correctness: As stated above, the empirical evaluation is performed on a simplified version of the proposed algorithms. It is not clear to me why the full Urejectron or Rejectron algorithms are never used in the evaluation?

Clarity: The paper is written well; particularly the introduction does a very good job in making the reader familiar with the problem statement and the main contributions of the paper.

Relation to Prior Work: Most relevant prior work is cited; however, work on "unrestriced adversarial examples" (e.g. https://openreview.net/forum?id=Sye_OgHFwH) should be discussed additionally. I am not very familiar with the field of selective classification, so I might miss relevant work from this field.

Reproducibility: Yes

Additional Feedback: Generally, I am willing to increase my score if my major criticisms raised under Weaknesses were to be addressed. ### Post-rebuttal Thanks for your kind and extensive response to my review. Most of my main points have been addressed. However, I still miss an actual evaluation of the proposed methods (at least on toy data), An evaluation of a very much ablated version of the Urejectron and leaving an actual evaluation to future work as stated in the response is not sufficient in my opinion, particularly as it should not be a major effort on toy data. I thus keep my score of 5, even though I otherwise share much of the positive feedback of the other reviewers.


Review 2

Summary and Contributions: Short summary: ============== This paper considers the learning setting where (a) the learner is tested on a distribution that is different from training distribution but additionally, (b) the learner can output what is called a selective classifier (i.e., a classifier that can either output a label or just say "don't know" and abstain from predicting anything) and even further (c) the learner is given some unlabeled data from the test distribution. In this setting, the paper provides two algorithms with novel types of guarantees. More details ============= Specifically, the paper proposes two learning models: a) "The generalization setting" where the learner is given data from a training distribution (P) and unlabeled data from a test distribution (Q). The goal of the learner is to make sure that "rej_P + err_Q" is small. Here, rej_P is the fraction of the P distribution that falls under the "rejection set", and err_Q is the fraction of the Q distribution on which the learner mislabels (which would NOT include the rejection set). b) "The transductive setting" where Q=P. Here, the learner first chooses an h using data from P. Then a white-box adversary that knows h and can see some test data Z from P creates arbitrary test data \tilde{X}. The learner can look at \tilde{X} to come up with a rejection set such that "rej_{Z} + err_{\tilde{X}}" is small. Under these settings, the paper presents -- an supervised learning algorithm Rejectron that outputs a hypothesis + the acceptance set. -- and an unsupervised learning algorithm Urejectron that outputs only an acceptance set (and has access to only unlabeled data from P) For both these algorithms the paper demonstrates -- computational efficiency of the algorithms assuming access to an appropriate ERM oracle -- upper bounds on the above guarantees (in terms of the VC dimension). -- Since these bounds grow as \sqrt{d/n} unlike standard VC dim bounds (d/n), the paper also presents a matching lower bound to show that this is indeed tight. Finally, the paper presents some experiments running a practical version of the unsupervised algorithm on the EMNIST dataset for two different classifiers (Neural network & Random forest) and two different test distributions (natural and adversarial). These experiments are to be considered as a proof of concept.

Strengths: This paper was an exciting and enjoyable read! I've many positive comments and honestly nothing negative to say (maybe some suggestions at best). 1. The problem of being robust to adversarial shifts in the distribution is an important problem, and it is even more important and interesting to get a learning-theoretic grounding for it. Admittedly, unlike most practical settings, this paper works with a selective classifier. While the selective classifier might seem restrictive, such classifiers are interesting in their own right as ultimately, a "don't know" is always better than a misclassification. Additionally, it might also seem restrictive that the model uses unlabeled data, but the problem becomes much harder without such an assumption. Overall, even though the paper makes some seemingly restrictive assumptions, these assumptions make sense as they help us provide guarantees without making the problem trivial. 2. I'd also like to note that both the high level approach and the low level details are novel here. At a high level, the idea of using a selective classifier together with formulating the goal as a sum of rej_{train dist} + err_{test dist} is novel. This also gives way for a neat, formal treatment of the algorithms. At the low level, the algorithms are novel, yet simple to understand and natural (indeed they resemble a lot of standard disagreement-based algorithms common to KWIK-learning, mistake-bound learning, active learning, co-training etc.,). The proofs are simple to understand. It's interesting that the sample complexity captures a tradeoff between the two goals of minimizing error and minimizing rejection rate. 3. The paper also leaves open the possibility for quite a lot of follow-up work since the framework is general. One could imagine trying to come up with more sample-efficient/computationally-efficient algorithms under a variety of other assumptions within this framework. Overall I think that this is a valuable foundational paper in the theory of learning to be robust to arbitrary adversaries.

Weaknesses: I don't think I have anything significant to criticize in the paper.

Correctness: Yes, they seem correct. However, I've not checked the appendix proofs in detail.

Clarity: The paper is well-written, concise yet clear. The introduction provided a good idea of what's coming ahead through an illustrative toy example. The main paper summarizes the proof sketch well. 4. I just had a few suggestions here. My main suggestion is with the presentation of the transductive setting. First, I think it'd be useful to write this up into a formal definition (like PQ-learning). This would help the reader understand the exact kind of adversary this setting deals with. I had a little bit of confusion here partly because of the wording that "the adversary modified any number of examples from z to create \tilde{x}." 4.1 Does every example in \tilde{x} have to somehow map to some example in z? My understanding is that it doesn't have to. If not, could you clarify? 4.2 Why does this setting work with a finite z instead of P itself? Is it just to keep things consistent (i.e., the adversarial set is finite, and so should the non-adversarial set be)? Or does something break if we replace the set z with the distribution P? 5. Also, it took me a while to convince myself as to why the transductive problem setting is not trivial given that the learner can see all the unlabeled, arbitrary, adversarial test data \tilde{x} (unlike in the previous setting). It might be worth adding some toy illustration to appreciate this setup. 6. In the experiments section, are the "imperceptible modifications" in line 267, adversarial? i.e., are these pixels deliberately changed to do something, or are they just haphazardly changed? (if it's the latter it would be good to be explicit about it since this phrase usually co-occurs with adversarial perturbations). Very minor suggestions: 7. In the initial description of Urejectron, it'd be helpful to highlight that unlike Rejectron, this algorithm does not output a hypothesis, rather only the acceptance set (and note this again in line 238). 8. I'd personally use some symbol other than a question mark for "abstain" since it sometimes breaks the flow for me. But this is just a personal preference. 9. In L172: I'd also add what the adversary does *not* have access to e.g., I presume, the training data? Also, I didn't quite get what the "for simplicity" is supposed to indicate. Does it mean, the paper will later make this specific setting more general? (was it foreshadowing the transductive setting?) Or that the discussion easily extends to more general adversaries? (It doesn't I think?) Or does it mean, one could imagine more complicated adversaries, but this paper will only focus on this sort of an adversary? 10. In Line 176: I initially misunderstood rej_x to be the anaog for rej_Q which didn't make sense. So this line could be rephrased.

Relation to Prior Work: Yes. The paper describes existing theoretical results in selective classification noting why they don't apply in this setting. The paper is also honest about the fact that existing algorithms fail in the considered setting only because they don't assume access to unlabeled data. 11. In L159, it'd be nice to be clearer regarding "as opposed to... we consider arbitrary test examples". This seems to (wrongly) suggest that the notion of arbitrary test examples is neither weaker nor stronger than the notion of adversarial examples in deep learning literature. However, the notion of arbitrary test examples here does capture (and is stronger than) the latter, right? If it is right to say that the notion of adversarial examples is subsumed by "the arbitrary adversary", it might also be worth outlining/distinguishing work in adversarial examples that uses unlabeled data. e.g., https://papers.nips.cc/paper/9298-unlabeled-data-improves-adversarial-robustness.pdf, https://arxiv.org/abs/1905.13725, https://papers.nips.cc/paper/8792-robustness-to-adversarial-perturbations-in-learning-from-incomplete-data.pdf, https://arxiv.org/abs/1906.00555 (I don't need any clarification regarding these papers; just a suggestion for what could be added in the related work section.) Thanks for the paper!

Reproducibility: Yes

Additional Feedback: Updates after response: Thank you for responding to my questions. Although I'm not increasing the score, I would continue to argue for acceptance of this paper since it has strong theoretical value and novelty.


Review 3

Summary and Contributions: The paper proposes the setting where the learning algorithm gets iid training data from a distribution P and gets samples from a different distribution Q, and learns a classifier that can abstain from predicting on some examples. In the realizable case, and assuming the target concept class C has finite VC dimension and given an Empirical Risk Minimizer (ERM) for C, the paper gives an algorithm that learns a model that has a low abstention rate with respect to P and a low generalization error with respect to Q. It also considers the transductive and agnostic settings.

Strengths: + The direction is novel. The paper proposes an interesting novel setting, and can potentially open up an interesting and useful research direction. + The results are also novel (since no prior guarantees were known even for simple classes of functions). They are also quite powerful (several bounds nearly match the lower bounds). + The results are based on algorithms that are quite natural and can be practical (the assumption of an ERM can be approximately achieved for quite rich application settings).

Weaknesses: - The given setup does not seem to match some practical considerations. For example, it may be the case that the classifier has a large abstention rate with respect to Q. Are there ways to improve this with some assumptions on Q? When the statistical distance between P and Q is small, the current guarantee implies a small abstention rate with respect to Q. Can this be generalized/improved? ============after response====================== I've read all the reviews and the response. I would like to thank the authors for the points, and remain positive about the paper.

Correctness: correct

Clarity: Yes

Relation to Prior Work: yes

Reproducibility: Yes

Additional Feedback:

[Author Response · NeurIPS 2020]

The thorough reviews asked numerous questions and suggested great clarifications/improvements. Due to space limitations, we are not able to respond to all of them, but we will incorporate all the feedback into the next version.

**Reviewer 1.** You make an excellent point that the skin-tone face example is a poor motivating example since one should collect a diverse set of labeled faces for training. We will give this as an example of how *not* to use our algorithm, and we are so grateful that you identified this issue. We will also note that research on PQ algorithms may help raise awareness that P and Q may differ. We will switch to the following motivating natural example:

> One wishes to provide a service classifying medical scans as normal or abnormal. Training data consists of a set of volunteered examples hand-labeled by multiple radiologists over a period of time. Test examples are to be classified in large daily batches. A concerning distribution shift may occur due a new disease, e.g. COVID-19 scans show "the presence of bilateral nodular and peripheral ground glass opacities" which may not occur in labeled training data [Sawani, 2020]. If there aren't enough unlabeled test examples at classification time, periodic rejections may be useful *in hindsight*, e.g., it could be immensely helpful if a machine can recognize a new disease (or a problematic change in the scanning pipeline) at the end of a week.

About the question of when one cannot simply label a sample of test examples: note that large-scale classification services may have millions of test examples per hour, but it may be impractical or even unethical for human annotators to label them. For instance, privacy policies may prohibit random private Twitter or SnapChat messages from being classified by humans, though one may have curated messages (e.g., public tweets) at train time. One may simply not know how representative the training examples are of the private messages, and one would prefer to abstain then misclassify at test (deploy) time.

About the concern of whether or not the adversary can choose examples based on $S$, in fact, in our model $S$ is a given deterministic function of the training data and unlabeled test examples provided by the adversary. Since the adversary is all-knowing and all-powerful (e.g., knows the training data), they can determine exactly what $S$ will be based on their chosen test set. Hence $S$ is effectively known. Forcing the adversary to provide a batch of test examples avoids the need for an "arms race" as the classifier always wins the game.

In our empirical evaluation, the simple "baseline" worked really well on our toy example, illustrating our fundamental contribution: *PQ learning is possible but provably requires unlabeled test data*. That is why we did not implement any of our more advanced algorithms, thus their value remains theoretical at this point. Future work evaluating and designing new PQ algorithms on real-world datasets is indeed of interest.

Finally, thank you for referring us to the work on "unrestricted adversarial examples", we will add this to the related work in the updated version.

**Reviewer 2.** Thank you for the enthusiasm and presentation suggestions. We will try to clarify the transductive setting. As you suggest, one may think of a one-to-one map between examples in $\tilde{x}$ and z: one way to think about it is that every example in $\tilde{x}$ is the result of a manipulation of some example in z. To see why our transductive guarantees are about $z$ and not $P$, consider a case where $P = Q$ is uniform on a huge set $X$. It would be unreasonable to reject the entire test set $\tilde{x}$ (and thus achieve zero test error), though if this is all you rejected you would have a negligible rejection rate on $P$. Regarding L159, It is true that the notion of arbitrary test examples is more general than the notion of adversarial examples with some specific perturbation set (e.g. $\ell_p$ perturbations). However, in this paper we do selective classification, which is unlike work on adversarial examples where a prediction is usually always required (even on perturbations). Also, thank you for referring us to the line of work on adversarial examples that uses unlabeled data, we will cite these papers in the updated version. We will replace the ? symbol with a different symbol, and we will address your many other questions though space prohibits us from commenting on them here.

**Reviewer 3.** Thank you as well for your enthusiasm and suggestions. As you suggest, the rejection rate bounds can be improved beyond statistical distance. In Appendix B, we give an example where a tighter bound of 0.1 can be shown while the statistical distance is 0.91, and Equation (2) also gives a refinement. We agree that it would be especially interesting to consider better bounds specific to the given concept class $C$.

As the reviewers suggest, this work is clearly not the final word on learning with arbitrary adversarial examples but we hope the theoretical results on the possibility of PQ learning inspire people to work in this area.

# References

Jina Sawani. How does covid 19 appear in the lungs? *U of M Health Blog*, March 2 2020.


[Meta-Review · NeurIPS 2020]

The paper considers learning a binary classifier in a setting where training and test examples can be from arbitrarily different distributions. The authors approach this problem by giving a selective classification algorithm — which returns a classifier and a subset on which the classifier abstains from assigning a label — while incurring few abstentions and few misclassification errors. Authors show that the proposed algorithm achieves optimal guarantees for classes with bounded VC dimension. This is an exciting contribution and a very timely one given the growing interest in robust machine learning and a need to better understand transfer learning. The paper is written clearly, and the results and insights in the paper are compelling. Overall, a good paper. Accept!